# KODA: Contrastive Representation Comparison and Alignment for Vision-Language Foundation Models

Youqi Wu [1]   Mohammad Jalali [1]   Farzan Farnia [1]

## Abstract

Vision-language foundation models such as CLIP and SigLIP provide widely used representations for multimodal learning systems. While these models are typically compared through downstream performance, such evaluations often do not explain how their representations differ structurally. In this work, we study this problem through the task of *Contrastive Embedding Clustering*: identifying sample subsets that are weakly clustered under one representation but strongly clustered under another. We propose *Kernel Optimization for Discrepancy Analysis (KODA)*, a kernel-based framework for contrastive representation comparison and alignment. KODA constructs unified multimodal kernels through modality-wise kernel composition and formulates discrepancy discovery as a constrained optimization problem that searches for coherent structures in one representation while suppressing coherence in a reference representation. This yields interpretable discrepancy directions associated with specific sample subsets and modality interactions. To scale KODA to large vision-language datasets, we develop randomized low-dimensional approximations of joint kernels using random projections, including Random Fourier Features for shift-invariant kernels. Empirically, KODA identifies consistent and interpretable discrepancy structures across vision-language representations and provides sample subsets for representation alignment. The code is available at https://github.com/yokiwuuu/KODA.

---

[1]Department of Computer Science and Engineering, The Chinese University of Hong Kong. Correspondence to: Youqi Wu <yqwu24@cse.cuhk.edu.hk>, Mohammad Jalali <mjalali24@cse.cuhk.edu.hk>, Farzan Farnia <farnia@cse.cuhk.edu.hk>.

*Proceedings of the 43rd International Conference on Machine Learning*, Seoul, South Korea. PMLR 306, 2026. Copyright 2026 by the author(s).

## 1. Introduction

Multi-modal embedding models have become a central component of modern machine learning systems, enabling joint representation of images, text, and other modalities within a shared semantic space. Contrastive vision–language embeddings such as CLIP (Radford et al., 2021) and related paradigms including ALIGN (Jia et al., 2021), BLIP (Li et al., 2022), and BLIP-2 (Li et al., 2023) have demonstrated strong performance in cross-modal retrieval and transfer, and they are now widely used as fixed representation interfaces in larger pipelines. As a result, many vision–language foundation models and public variants coexist, differing in architecture, training objectives, and data sources (Cherti et al., 2023). This diversity motivates principled methods for comparing representations and characterizing *how* they differ in the structure they induce on data, beyond reporting aggregate downstream metrics.

Most existing comparisons of multi-modal representations rely on downstream task performance, such as retrieval accuracy or zero-shot classification on benchmark datasets. While effective for ranking models, such evaluations provide limited insight into how representations organize data. In vision–language settings, two models may achieve similar overall accuracy yet induce different groupings of image–text pairs, for example by emphasizing different semantic attributes, compositional patterns, or rare concepts. Identifying such fine-grained differences can support interpretability-oriented workflows, including targeted data curation, model selection, and representation alignment.

A recent line of work on interpretable embedding comparison has begun to study sample groups that are organized differently by two representations. In particular, the SPEC framework by Jalali et al. (2025a) compares two embeddings by constructing kernel similarity matrices on a shared reference dataset and analyzing the eigendecomposition of their kernel-difference matrix. This setting motivates the task we call *Contrastive Embedding Clustering*: identifying sample subsets that are weakly clustered under one representation but strongly clustered under another. However, the kernel-difference construction in SPEC does not explicitly enforce this asymmetric objective; its eigendirections may reflect structure present in both embeddings or aggregate

## Overview of KODA Embedding Comparison:

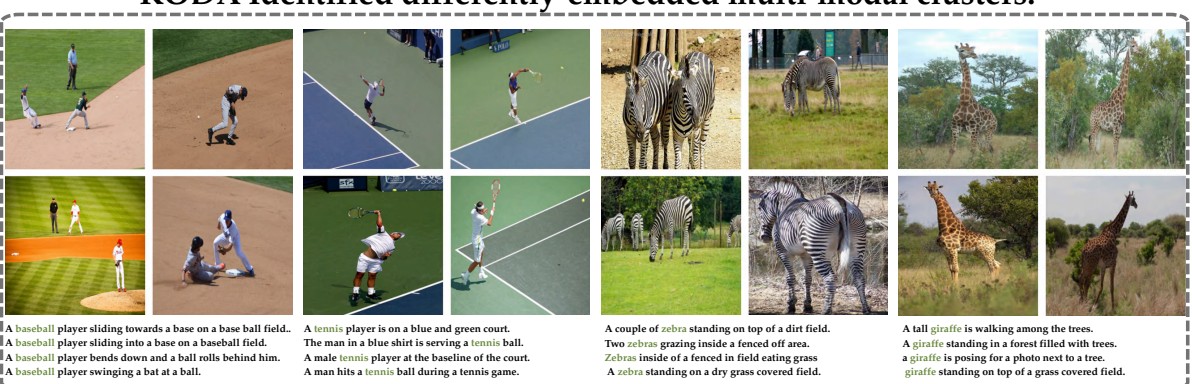

*Figure 1.* Overview of KODA for *Contrastive Embedding Clustering*, which aims to discover sample clusters that are represented differently by two embeddings. We show KODA-identified contrastive clusters for BLIP and CLIP embeddings on the MS-COCO dataset.

several effects, rather than isolating subsets that are weakly grouped with respect to a specified reference embedding.

In this work, we formulate Contrastive Embedding Clustering as a constrained optimization problem. Given kernel similarity matrices $K_A$ and $K_B$ induced by two representations, KODA seeks components that are strongly grouped under one representation while explicitly constrained to be weakly grouped under the other. We introduce *Kernel Optimization for Discrepancy Analysis* (KODA), which solves

$$\max_{x \in \mathbb{R}^n} \quad x^\top K_A x$$
$$\text{subject to} \quad x^\top K_B x \leq \epsilon, \quad (1)$$
$$\|x\|_2 = 1.$$

To identify multiple discrepancy modes, we solve a sequence of such problems with orthogonality constraints between the current and previous solutions. Although this optimization problem is non-convex, we show that it admits an efficient solution via structured spectral computations.

We further extend KODA to the comparison of multi-modal representations by constructing unified kernels through modality-wise kernel multiplication. This product-kernel

formulation enables direct comparison of representations over paired data, such as image–text samples, within the same constrained framework. However, exact covariance-operator implementations become infeasible in multi-modal settings because effective feature dimensions can scale multiplicatively across modalities.

To enable scalable computation, we develop approximations based on random projections and Random Fourier Features (Rahimi & Recht, 2007), which reduce the effective dimensionality of the joint kernel feature map while preserving approximation guarantees. We evaluate KODA on widely used vision–language representation models, including CLIP (Radford et al., 2021), ALIGN (Jia et al., 2021), and BLIP-style models (Li et al., 2022; 2023), using standard benchmarks such as MS-COCO (Lin et al., 2014). Our experiments show that KODA identifies consistent discrepancy structures, improves over kernel-difference baselines in finding multiple discrepancy modes, and provides sample subsets that can be used for representation alignment.

## 2. Related Work

**Multi-modal embedding models.** Vision–language embedding models are now a standard component in cross-modal

retrieval, zero-shot recognition, and multi-modal generative AI pipelines. Representative approaches include CLIP (Radford et al., 2021), ALIGN (Jia et al., 2021), and BLIP / BLIP-2 (Li et al., 2022; 2023), alongside scaling studies and public training efforts that broaden the space of available variants (Cherti et al., 2023). SigLIP (Zhai et al., 2023) and SigLip 2 (Tschannen et al., 2025) propose an alternative pre-training objective based on a sigmoid loss. While these models are commonly compared via downstream benchmarks, such evaluations often provide limited insight into how two embeddings organize the *same* paired data.

**Explainability in representation learning.** A broad line of research develops tools to interpret learned representations directly, often by linking internal directions or units to human-understandable concepts. Network Dissection (Bau et al., 2017) quantifies interpretability of visual representations by measuring alignment between hidden units and semantic concepts. Concept-based explanation methods use examples of a concept to define directions in representation space: TCAV (Kim et al., 2018) measures concept sensitivity via directional derivatives, while ACE (Ghorbani et al., 2019) automatically extracts concepts and evaluates their importance, reducing reliance on manually specified concept sets. Concept Bottleneck Models (Koh et al., 2020) further emphasize interpretability by explicitly structuring representations around a set of supervised concepts and enabling interventions on that representation. More recently, Gong et al. (2025b) improve the visual interpretability of CLIP through unsupervised adversarial fine-tuning with norm regularization, showing gains through feature-attribution and network-dissection analyses. These works motivate analyses that identify *which parts of a dataset* correspond to salient representational structure, but they do not directly target the embeddings' discrepancy discovery.

**Kernel-based methods for feature representations.** Kernel methods provide a flexible way to compare, align, fuse, and evaluate learned feature representations through pairwise similarity structure rather than only downstream task accuracy. Recent work has used kernel matrices to explain differences between embedding spaces and align their induced cluster structures (Jalali et al., 2025a), and to improve vision-language representations by aligning CLIP visual embeddings with stronger vision-centric embeddings such as DINOv2 (Gong et al., 2025a). Complementarily, kernel product feature maps and maximum kernel entropy methods have been used to fuse and recover embeddings while preserving pairwise similarity information (Wu et al., 2025; Wu & Farnia, 2026). Kernel-based scores have also become central to generative-model evaluation, beginning with MMD-based comparison (Gretton et al., 2012) and the Kernel Inception Distance (Bińkowski et al., 2018; Wang et al., 2025), and extending to entropy- and spectrum-based measures for diversity, novelty, and prompt-aware evaluation,

including RKE (Jalali et al., 2023), Vendi (Friedman & Dieng, 2023; Ospanov et al., 2024), KEN (Zhang et al., 2024; 2025), and conditional kernel entropy (Jalali et al., 2025b; Ospanov et al., 2025; Jalali et al., 2026). Kernel-based uses of embeddings have also been explored for model evaluation, selection, and mixture construction (Stein et al., 2023; Hu et al., 2025b; Rezaei et al., 2025; Hu et al., 2025a; Jafari & Farnia, 2026). These methods motivate viewing embeddings and foundation models not only as feature vectors for prediction, but also as kernel-induced representations whose geometry can be systematically compared and evaluated.

## 3. Preliminaries

### 3.1. Embedding maps & comparison setting

Let $\mathcal{X}$ denote an input space and let $\psi : \mathcal{X} \to \mathcal{S}$ be an embedding map into a representation space $\mathcal{S}$ (typically Euclidean). We consider two embedding maps

$$\psi_1 : \mathcal{X} \to \mathcal{S}_1, \qquad \psi_2 : \mathcal{X} \to \mathcal{S}_2,$$

which may have different output dimensions and thus induce different similarity structures on the same inputs. We assume access to a reference dataset $\{x_i\}_{i=1}^n \subset \mathcal{X}$ sampled from an underlying distribution. Our goal is to compare $\psi_1$ and $\psi_2$ through the geometry they induce on this reference set, without relying on labeled downstream tasks.

### 3.2. Kernel functions & kernel-induced quadratic forms

A kernel function $k : \mathcal{X} \times \mathcal{X} \to \mathbb{R}$ assigns a similarity score and admits a feature map $\phi : \mathcal{X} \to \mathcal{H}$ into a (possibly infinite-dimensional) Hilbert space $\mathcal{H}$ such that

$$k(x, x') = \langle \phi(x), \phi(x') \rangle_{\mathcal{H}}.$$

Given samples $x_1, \ldots, x_n$, the associated kernel matrix $K \in \mathbb{R}^{n \times n}$ is

$$K_{ij} = k(x_i, x_j), \tag{2}$$

and is positive semidefinite. Common normalized examples include the cosine kernel $k_{\cos}(u, v) = \frac{u^\top v}{\|u\|_2 \|v\|_2}$ (for nonzero $u, v$) and the Gaussian (RBF) kernel $k_{\mathrm{rbf}}(u, v) = \exp\left( -\|u - v\|_2^2 / (2\sigma^2) \right)$, both satisfying $k(x, x) = 1$.

For any $v \in \mathbb{R}^n$ with $\|v\|_2 = 1$, the quadratic form

$$v^\top K v \tag{3}$$

is the Rayleigh quotient of $K$ at $v$, and hence lies in $[\lambda_{\min}(K), \lambda_{\max}(K)]$ and is maximized by a top eigenvector of $K$. Moreover, when $\phi$ is finite-dimensional and $K = \Phi \Phi^\top$ with $\Phi_{i:} = \phi(x_i)^\top$, we have the identity

$$v^\top K v = \|\Phi^\top v\|_2^2 = \left\| \sum_{i=1}^n v_i \, \phi(x_i) \right\|_2^2, \tag{4}$$

which will be useful when interpreting and optimizing kernel-based criteria on the reference set. When $\phi(x) \in \mathbb{R}^d$,

the empirical covariance matrix (operator) is

$$C_X := \frac{1}{n}\Phi^\top\Phi = \frac{1}{n}\sum_{i=1}^n \phi(x_i)\phi(x_i)^\top \in \mathbb{R}^{d\times d}. \quad (5)$$

The matrices $K/n$ and $C_X$ share the same non-zero eigenvalues (including multiplicities), since they are products of $\Phi$ and $\Phi^\top$ in opposite orders.

### 3.3. Shift-invariant kernels & random Fourier features

To scale kernel computations, we use random Fourier features (RFF) (Rahimi & Recht, 2007; Sutherland & Schneider, 2015) for shift-invariant kernels on $\mathbb{R}^d$. Consider kernels of the form $k(x, x') = \kappa(x-x')$, where $\kappa$ is continuous and positive definite. By Bochner's theorem, there exists a non-negative finite measure (taking non-negative values) $\widehat{\kappa}$, which is the Fourier transform of $\kappa$, such that

$$\kappa(\delta) = \int_{\mathbb{R}^d} \exp\bigl(\mathrm{i}\omega^\top\delta\bigr)\widehat{\kappa}(\omega)\mathrm{d}\omega. \quad (6)$$

For a normalized kernel with $\kappa(0) = 1$, $\widehat{\kappa}$ will be a probability measure. We then sample $\omega_1, \ldots, \omega_m \overset{\text{i.i.d.}}{\sim} \widehat{\kappa}$ and define the RFF proxy feature map $\varphi_r : \mathcal{X} \to \mathbb{R}^{2r}$ as

$$\varphi_r(x) = \frac{1}{\sqrt{r}}\bigl[\cos(\omega_1^\top x), \sin(\omega_1^\top x), ., \cos(\omega_r^\top x), \sin(\omega_r^\top x)\bigr]. \quad (7)$$

A direct calculation yields $\mathbb{E}[\langle\varphi_r(x), \varphi_r(x')\rangle] = k(x, x')$, where the expectation is over the sampled frequency vectors. Given $\{x_i\}_{i=1}^n$, we form the approximate kernel matrix $\widetilde{K}$ by $\widetilde{K}_{ij} = \langle\varphi_r(x_i), \varphi_r(x_j)\rangle$, enabling computation of kernel quadratic forms and spectral quantities using an explicit feature representation.

## 4. KODA: Optimization-based discrepancy identification in kernel matrices

We formalize the comparison problem as a *Contrastive Embedding Clustering* task. Given two embeddings $A$ and $B$ evaluated on the same reference set $\{x_1, \ldots, x_n\}$, the goal is to identify subsets or signed directions over the reference samples that form a coherent cluster under one embedding but not under the other. In this sense, the task is not merely to measure a global discrepancy between embeddings, but to localize it by extracting directions in the reference set where the two kernel similarity geometries disagree.

For two embeddings $A$ and $B$, we are given their normalized kernel similarity matrices $K_A, K_B \in \mathbb{R}^{n\times n}$ constructed on the same reference set $\{x_1, \ldots, x_n\}$, where $\frac{1}{n}K_A \succeq 0$ and $\frac{1}{n}K_B \succeq 0$ are PSD and unit-trace. We develop an optimization formulation for Contrastive Embedding Clustering: extracting directions on the reference set along which the structure induced by one embedding is strongly clustered while the structure by the other is weakly clustered.

### 4.1. Constrained quadratic programming for kernel-based embedding comparison

For a target level $\epsilon > 0$, we consider the following quadratically constrained program for Contrastive Embedding Clustering. The constraint enforces weak clusterability under $K_B$, while the objective searches for a direction that is maximally clustered under $K_A$:

$$\begin{aligned}\max_{x\in\mathbb{R}^n} \quad & x^\top K_A\, x \\ \text{s.t.} \quad & x^\top K_B\, x \le \epsilon, \\ & \|x\|_2^2 = 1.\end{aligned} \quad (8)$$

Let $x^\star$ denote an optimizer. The constraint $x^\top K_B, x \le \epsilon$ enforces that the discrepancy direction has limited energy under the second embedding, while the objective selects the direction with the strongest similarity concentration under the first embedding. Thus, $x^\star$ identifies a contrastive cluster direction: a set-level pattern that is salient in embedding $A$ but suppressed, diffuse, or absent in embedding $B$.

Although the optimization problem (8) is a non-convex optimization task (maximizing a convex objective function), its optimizers admit an eigenvector-based characterization as revealed by KKT conditions.

**Proposition 4.1** (Eigenvector form of an optimizer)**.** *Assume (8) is feasible and there exists $\bar{x}$ with $\|\bar{x}\|_2 = 1$ and $\bar{x}^\top K_B\, \bar{x} < \epsilon$. Then, there exist scalars $\lambda^\star \ge 0$ and $\nu^\star \in \mathbb{R}$ such that any optimizer $x^\star$ with unit-norm ($\|x^\star\|_2 = 1$) satisfies*

$$(K_A - \lambda^\star K_B)\, x^\star = \nu^\star x^\star, \quad (9)$$

$$\lambda^\star\bigl(x^{\star\top} K_B\, x^\star - \epsilon\bigr) = 0. \quad (10)$$

*Proof.* We present the proof in the Appendix. $\square$

**Searching over $\lambda$.** Proposition 4.1 motivates a one-dimensional search over $\lambda \ge 0$: for each $\lambda$, compute a leading eigenvector $x_\lambda$ of $K_A - \lambda K_B$ (normalized to $\|x_\lambda\|_2 = 1$) and evaluate $g(\lambda) := x_\lambda^\top K_B\, x_\lambda$. We then select a $\lambda$ that yields $g(\lambda) \le \epsilon$ and maximizes $x_\lambda^\top K_A\, x_\lambda$ among such candidates (up to numerical tolerance).

### 4.2. Iterative extraction of discrepancy directions via KODA

A single solution of (8) yields one contrastive cluster direction. KODA solves the Contrastive Embedding Clustering task by extracting such directions, iteratively solving (8) while enforcing orthogonality to the found directions.

Let $x_1, \ldots, x_{t-1}$ be previously extracted unit vectors, and let $U_{t-1} \in \mathbb{R}^{n\times(t-1)}$ have orthonormal columns spanning $\text{span}\{x_1, \ldots, x_{t-1}\}$. Define the orthogonal projector

$$P_{t-1} := I - U_{t-1}U_{t-1}^\top. \quad (11)$$

At iteration $t$, we solve

$$\max_{x \in \mathbb{R}^n} \quad x^\top K_A \, x$$
$$\text{s.t.} \quad x^\top K_B \, x \le \epsilon,$$
$$\|x\|_2 = 1,$$
$$U_{t-1}^\top x = 0. \tag{12}$$

**Proposition 4.2.** *The optimization problem* (12) *is equivalent to*

$$\max_{x \in \mathbb{R}^n} \quad x^\top (P_{t-1} K_A \, P_{t-1}) x$$
$$s.t. \quad x^\top (P_{t-1} K_B \, P_{t-1}) x \le \epsilon,$$
$$\|x\|_2 = 1, \tag{13}$$

*and every optimizer of* (13) *satisfies* $U_{t-1}^\top x = 0$.

*Proof.* We present the proof in the Appendix. $\qquad\square$

Algorithm 1 summarizes the steps in KODA. At each iteration, we work with the projected matrices $A = P K_A \, P$ and $B = P K_B \, P$ from (13) and perform a 1D search over $\lambda$ via repeated leading-eigenvector computations.

## 4.3. Scalable principal-eigenvector computation via covariance blocks

KODA requires repeated computation of a principal eigenvector of matrices of the form $K_A - \lambda K_B \in \mathbb{R}^{n \times n}$. For large reference sets (e.g., $n \gtrsim 20000$), this step can be a computational bottleneck in the dense-kernel regime: even iterative eigensolvers (Lanczos/power iteration) rely on repeated matrix–vector products, each costing $\Theta(n^2)$ time (and $\Theta(n^2)$ memory if $K_A, K_B$ are explicitly formed). When the kernels admit explicit feature representations with feature dimensions $d_1, d_2 \ll n$ (e.g., via random features), the same principal-eigenvector computation can be reduced to an eigenproblem of dimension $d_1 + d_2$.

Assume the kernel matrices factor as

$$K_A = \Phi_A \Phi_A^\top, \qquad K_B = \Phi_B \Phi_B^\top, \tag{14}$$

with $\Phi_A \in \mathbb{R}^{n \times d_1}$ and $\Phi_B \in \mathbb{R}^{n \times d_2}$. Let $\Phi := [\Phi_A \ \Phi_B] \in \mathbb{R}^{n \times (d_1 + d_2)}$ and define the covariance blocks

$$C_{AA} := \Phi_A^\top \Phi_A, \qquad C_{AB} := \Phi_A^\top \Phi_B,$$
$$C_{BA} := \Phi_B^\top \Phi_A, \qquad C_{BB} := \Phi_B^\top \Phi_B, \tag{15}$$

so that $G := \Phi^\top \Phi = \begin{bmatrix} C_{AA} & C_{AB} \\ C_{BA} & C_{BB} \end{bmatrix}$.

**Proposition 4.3.** *For coefficient* $\lambda \ge 0$, *define block matrices* $S_\lambda := \mathrm{diag}(I_{d_1}, -\lambda I_{d_2})$ *and*

$$M_\lambda := S_\lambda G = \begin{bmatrix} C_{AA} & C_{AB} \\ -\lambda C_{BA} & -\lambda C_{BB} \end{bmatrix}. \tag{16}$$

*Let* $\eta_\lambda := \lambda_{\max}(K_A - \lambda K_B)$ *and let* $u_\lambda$ *be a (right) eigenvector of* $M_\lambda$ *associated with eigenvalue* $\eta_\lambda$. *Then,* $x_\lambda := \Phi u_\lambda$ *is a principal eigenvector of* $K_A - \lambda K_B$ *(after normalization).*

*Proof.* We present the proof in the Appendix. $\qquad\square$

**Discussion and computational implications.** Proposition 4.3 reduces the principal-eigenvector computation of the $n \times n$ matrix $K_A - \lambda K_B$ to an eigenproblem of size $(d_1 + d_2) \times (d_1 + d_2)$, which is independent of $n$. This is particularly beneficial when $d_1, d_2 \ll n$, as is typical with explicit feature maps or random features. In practice, we form the covariance blocks in (15) in $\mathcal{O}(n(d_1 + d_2)^2)$ time (or $\mathcal{O}(n(d_1 + d_2))$ time if $C_{ij}$ are accumulated online with a single pass) and then compute a dominant eigenvector of $M_\lambda$. The lifted vector $x_\lambda = \Phi u_\lambda$ can be obtained without forming any $n \times n$ matrix, and can be normalized to satisfy $\|x_\lambda\|_2 = 1$ before evaluating the constraint quantity $x_\lambda^\top K_B \, x_\lambda$ (which can likewise be computed via $\Phi_B$ as $x_\lambda^\top K_B \, x_\lambda = \|\Phi_B^\top x_\lambda\|_2^2$).

**Sample complexity of the covariance-block eigendirections.** To state a population sample-complexity guarantee, we use the *normalized* covariance blocks $\widehat{C}_{ij} := \frac{1}{n} \Phi_i^\top \Phi_j$ and $\widehat{G} := \frac{1}{n} \Phi^\top \Phi$. Let $z(x) := [\Phi_A(x); \Phi_B(x)] \in \mathbb{R}^{d_1 + d_2}$ denote the per-sample feature vector (the $i$th row of $\Phi$ is $z(x_i)^\top$). Assume $\|\Phi_A(x)\|_2 \le 1$ and $\|\Phi_B(x)\|_2 \le 1$ for all $x$ (equivalently, $\|z(x)\|_2^2 \le 2$). Define the population block covariance $G := \mathbb{E}[z(X) z(X)^\top]$ and $S_\lambda := \mathrm{diag}(I_{d_1}, -\lambda I_{d_2})$. Finally, define the symmetric matrices $B_\lambda := G^{1/2} S_\lambda G^{1/2}$ and $\widehat{B}_\lambda := \widehat{G}^{1/2} S_\lambda \widehat{G}^{1/2}$.

**Theorem 4.4.** *Consider the setting described above. Then, for every* $\lambda \ge 0$ *and* $\delta \in (0, 1)$, *the following holds with probability at least* $1 - \delta$,

$$\left\| \widehat{B}_\lambda - B_\lambda \right\|_2 \le 12 \, \|S_\lambda\|_2 \sqrt[4]{\frac{d_1 + d_2}{n}} \left( 1 + \sqrt{\log(1/\delta)} \right).$$

*Moreover, if* $B_\lambda$ *has eigengap* $\gamma_\lambda := \lambda_1(B_\lambda) - \lambda_2(B_\lambda) > 0$, *and* $v_1, \widehat{v}_1$ *are unit top eigenvectors of* $B_\lambda, \widehat{B}_\lambda$, *then*

$$\sin \angle(\widehat{v}_1, v_1) \le \frac{\|\widehat{B}_\lambda - B_\lambda\|_2}{\gamma_\lambda}. \tag{17}$$

*Proof.* We present the proof in the Appendix. $\qquad\square$

# 5. KODA for Multi-modal embeddings via product kernels and random features

We extend KODA to the comparison of multi-modal embeddings, where each reference item consists of paired observations from different modalities. We first introduce a

**Algorithm 1** KODA: Kernel Optimization for Discrepancy Analysis

---

**Require:** PSD kernels $K_A$, $K_B \in \mathbb{R}^{n \times n}$, threshold $\epsilon > 0$, number of directions $T$, tolerance $\tau$
**Ensure:** Directions $x_1, \ldots, x_T \in \mathbb{R}^n$
1: $U \leftarrow [\,]$ (null matrix)
2: **for** $t = 1, \ldots, T$ **do**
3:    $P \leftarrow I - UU^\top$
4:    $A \leftarrow PK_A P, \quad B \leftarrow PK_B P$
5:    Choose $\lambda_t \geq 0$ by a 1D search using: compute a leading eigenvector $x_\lambda$ of $A - \lambda B$ with $\|x_\lambda\|_2 = 1$, and evaluate $g(\lambda) = x_\lambda^\top B x_\lambda$
6:    Stop when $g(\lambda_t) \leq \epsilon + \tau$ and set $x_t \leftarrow x_{\lambda_t}$
7:    Orthonormalize: $U \leftarrow \mathrm{orth}([U \;\; x_t])$
8: **end for**
9: **Return** $\{x_t\}_{t=1}^T$

---

product-kernel formulation that induces a joint similarity matrix on paired samples. We then address the computational challenge posed by the tensor-product feature space of product kernels by proposing a joint random Fourier feature approximation, and establish a guarantee on the stability of the leading eigenspaces used by KODA.

### 5.1. Product-kernel formulation and tensor-product bottleneck

For each reference sample that paired is $z_i = (x_i, t_i)$, let

$$u_i = \psi_x(x_i) \in \mathbb{R}^{d_x}, \qquad v_i = \psi_t(t_i) \in \mathbb{R}^{d_t}.$$

We consider normalized shift-invariant kernels for each modality as follows where $\kappa_x(0) = \kappa_t(0) = 1$:

$$k_x(u, u') = \kappa_x(u - u'), \;\; k_t(v, v') = \kappa_t(v - v') \quad (18)$$

Then, we define the multi-modal product kernel as:

$$\begin{aligned} k\big((u, v), (u', v')\big) &= k_x(u, u')\, k_t(v, v') \\ &= \kappa_x(u - u')\, \kappa_t(v - v'). \end{aligned} \quad (19)$$

For a reference set $\{(u_i, v_i)\}_{i=1}^n$, the corresponding kernel matrix satisfies $K = K_x \odot K_t$ where

$$(K_x)_{ij} = k_x(u_i, u_j), \quad (K_t)_{ij} = k_t(v_i, v_j), \quad (20)$$

with $\odot$ denoting the Hadamard product. At the kernel level, KODA applies directly by replacing unimodal kernels in Section 4 with $K$.

The challenge arises in covariance-based implementations: the feature map associated with (19) is the tensor product

$$\phi(u, v) = \phi_x(u) \otimes \phi_t(v),$$

whose ambient dimension scales as $d_x d_t$. This renders covariance-space eigen-computations infeasible for standard multi-modal embeddings, motivating a low-dimensional kernel approximation.

### 5.2. Joint random Fourier features

Since $\kappa_x$ and $\kappa_t$ are continuous, real-valued, and shift-invariant, Bochner's theorem yields probability measures $\widehat{\kappa}_x$ and $\widehat{\kappa}_t$ such that

$$\kappa_x(\delta) = \int_{\mathbb{R}^{d_x}} \widehat{\kappa}_x(\omega_x) \cos(\omega_x^\top \delta)\, \mathrm{d}\omega_x,$$

$$\kappa_t(\zeta) = \int_{\mathbb{R}^{d_t}} \widehat{\kappa}_t(\omega_t) \cos(\omega_t^\top \zeta)\, \mathrm{d}\omega_t. \quad (21)$$

We sample $r$ frequency vectors independently for each modality:

$$\omega_{x,1}, \ldots, \omega_{x,r} \overset{\mathrm{iid}}{\sim} \widehat{\kappa}_x, \quad \omega_{t,1}, \ldots, \omega_{t,r} \overset{\mathrm{iid}}{\sim} \widehat{\kappa}_t,$$

and define the joint random Fourier feature map

$$\varphi(u, v) = \frac{1}{\sqrt{r}} \Big[ \cos(\omega_{x,1}^\top u + \omega_{t,1}^\top v), \sin(\omega_{x,1}^\top u + \omega_{t,1}^\top v),$$

$$\ldots, \cos(\omega_{x,r}^\top u + \omega_{t,r}^\top v), \sin(\omega_{x,r}^\top u + \omega_{t,r}^\top v) \Big] \in \mathbb{R}^{2r}. \quad (22)$$

Let $\Phi \in \mathbb{R}^{n \times 2r}$ contain rows $\Phi_{i:} = \varphi(u_i, v_i)^\top$. The approximate kernel matrix is

$$\widetilde{K} = \Phi\Phi^\top \quad \text{where} \quad \widetilde{K}_{ij} = \langle \varphi(u_i, v_i), \varphi(u_j, v_j) \rangle. \quad (23)$$

Next, we show a theoretical guarantee supporting scalable multi-modal KODA via the joint random Fourier feature implementation:

**Theorem 5.1.** *Let $K$ be defined in (20) and $\widetilde{K}$ in (23). Assume $|K_{ij}| \leq 1$ for all $i, j$. Then for any $\delta \in (0, 1)$, with probability at least $1 - \delta$,*

$$\Big\| \frac{1}{n}\widetilde{K} - \frac{1}{n}K \Big\|_F \;\leq\; \frac{2 + \sqrt{8\log(1/\delta)}}{\sqrt{r}} \quad (24)$$

*Moreover, for any $q$ with eigengap $\Delta_q(K) = \lambda_q(K) - \lambda_{q+1}(K) > 0$, letting $U$ and $\widetilde{U}$ denote the top-$q$ eigenspaces of $K$ and $\widetilde{K}$ respectively,*

$$\big\| \sin\Theta(\widetilde{U}, U) \big\|_F \;\leq\; \frac{\|\widetilde{K} - K\|_F}{\Delta_q(K)}. \quad (25)$$

*Proof.* We present the proof in the Appendix. $\qquad \square$

Theorem 5.1 provides justification for using $\widetilde{K} = \Phi\Phi^\top$ as a proxy for the true product kernel matrix $K$ in multi-modal KODA. In particular, $\widetilde{K}$ concentrates around $K$ at rate $r^{-1/2}$ in Frobenius norm, and the leading eigenspaces are stable whenever $\Delta_q(K)$ is non-negligible. This enables covariance/feature-space implementations whose complexity depends on the joint random-feature dimension $2r$, avoiding explicit tensor-product feature maps of size $d_x d_t$.

Top 6 DINOv2-dominant directions relative to CLIP identified by KODA

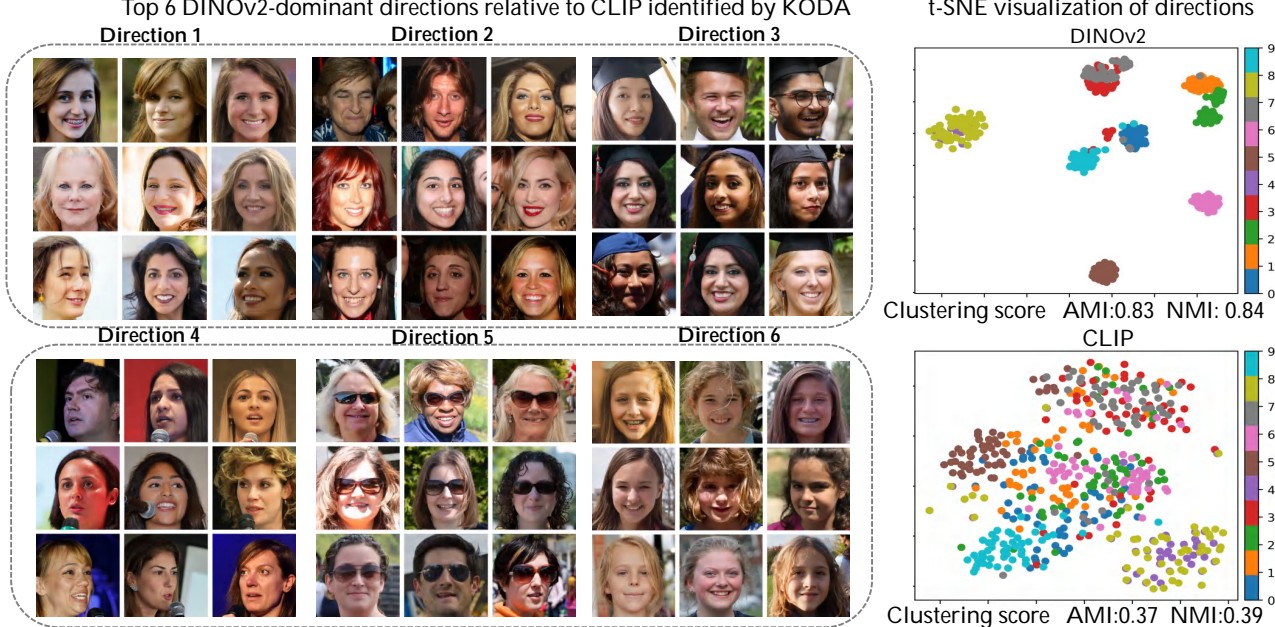

*Figure 2.* Left: Visualization of the top-6 discrepancy directions that are strongly grouped under DINOv2 while being weakly clustered under CLIP on the FFHQ dataset, discovered by KODA. Right: t-SNE visualization of Top-10 directions together with clustering scores.

## 6. Numerical Results

In this section, we evaluate KODA through two complementary tasks. The first task, *contrastive embedding clustering*, asks whether KODA can identify sample groups that are coherently clustered under one embedding but weakly clustered under another. The second task, *contrastive embedding alignment*, asks whether the discovered contrastive clusters can be used as actionable slices for targeted alignment between embeddings. Finally, we provide ablation studies on key design choices in KODA.

**Datasets.** We evaluate our method on a diverse collection of image-only and image–text datasets to assess discrepancy discovery under both unimodal and multimodal settings. For image-only experiments, we use AFHQ (Choi et al., 2020), FFHQ (Karras et al., 2019), and ImageNet (Deng et al., 2009). For multimodal experiments, we adopt standard image-caption datasets MSCOCO (Lin et al., 2014).

**Models.** For unimodal discrepancy analysis, we consider two widely adopted visual encoders, DINOv2 (Oquab et al., 2023) and CLIP (Radford et al., 2021). For multimodal experiments, we evaluate a diverse set of vision-language models, including BLIP (Li et al., 2022), CLIP (Radford et al., 2021), OpenCLIP (Ilharco et al., 2021), SigLIP (Zhai et al., 2023), and SigLIP2 (Tschannen et al., 2025).

**Implementation details.** All experiments are conducted using the covariance-operator formulation of KODA, with spectral computations solved via Cholesky decomposition.

We adopt Gaussian (RBF) kernels and kernel bandwidths are selected following prior work (Zhang et al., 2024) (Jalali et al., 2025a) to ensure comparable scaling across embeddings. Further implementation details are provided in C.1.

**Contrastive embedding clustering in unimodal encoders.** We begin with the contrastive embedding clustering task for two image encoders, DINOv2 and CLIP, on the AFHQ and FFHQ datasets. Figure 2 illustrates the dominant discrepancy directions identified by KODA that are strongly grouped under DINOv2 while being weakly clustered under CLIP. For visualization, we select 50 representative samples per direction and project their embeddings using t-SNE (Van der Maaten & Hinton, 2008) under each model. To quantify the identified directions, we run $k$-means on the corresponding embeddings with 10 times and report the averaged Adjusted Mutual Information (Vinh et al., 2009) (AMI) and Normalized Mutual Information (McDaid et al., 2013) (NMI) between the $k$-means labels and the KODA-discovered labels. The results on AFHQ datasets are provided in Figure 7 and Figure 8.

**Consistency with reference discrepancy structures.** We further examine whether the discrepancy directions identified by KODA align with semantic mismatches derived from representation similarity statistics. We use the ImageNet dog breeds dataset since it provides category labels that enable explicit verification. We derive ground-truth discrepancy labels based on aggregated similarity statistics. Without using any label information, KODA recovers domi-

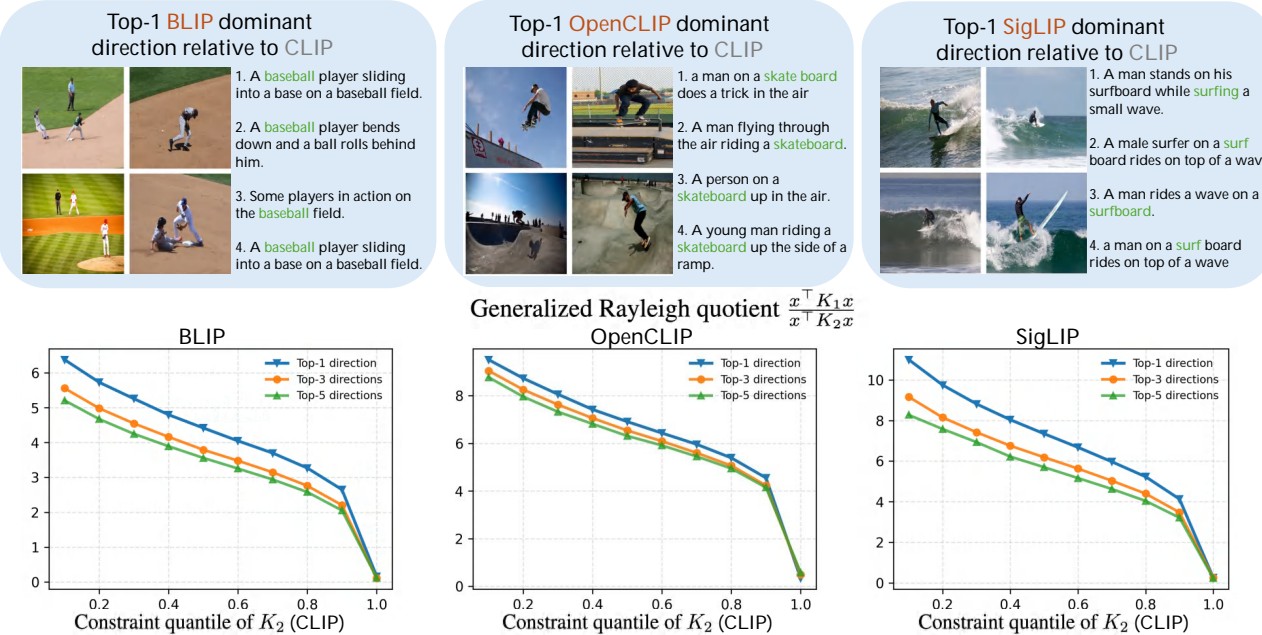

*Figure 3.* Multimodal discrepancy analysis on the MSCOCO dataset. **Top:** Representative image–caption pairs corresponding to the Top-1 discrepancy direction identified by KODA for different vision–language models relative to CLIP. **Bottom:** Generalized Rayleigh quotient of the identified discrepancy directions under varying constraint quantiles defined on the CLIP kernel.

nant discrepancy directions that closely correspond to these mismatched categories as shown in Figure 6. Additional results and visualizations are provided in Appendix C.2.

**Contrastive embedding clustering in vision–language models.** We next evaluate KODA on contrastive embedding clustering for paired image–text data. All multimodal experiments are conducted on MSCOCO using the joint image–text representation described in Section 5.2. We consider a set of widely used vision–language models, including BLIP, CLIP, OpenCLIP, SigLIP, and SigLIP2, and perform pairwise discrepancy analysis. As shown in Figure 3 (top), the resulting samples exhibit distinct multimodal patterns across different models by fixing CLIP as the reference model. Additional comparing results and visualization across different models are provided in Appendix C.4.

**Quantifying Directional Asymmetry via the Generalized Rayleigh Quotient.** We further quantify the strength of multimodal discrepancy directions using the generalized Rayleigh quotient $\frac{x^\top K_1 x}{x^\top K_2 x}$, where $K_1$ and $K_2$ are normalized RBF kernel matrices induced by the two embeddings. Since $x^\top K x$ measures how strongly direction $x$ is expressed under kernel $K$, larger quotient values indicate stronger directional asymmetry, i.e., directions emphasized by $K_1$ but suppressed by $K_2$. Following Eq. (7), the constraint parameter $\epsilon$ is set *implicitly* via a quantile $q \in \{0.1, 0.2, \ldots, 1.0\}$ of the eigenvalue distribution of $K_2$, where $\epsilon$ corresponds

to the $q$-quantile of $K_2$'s eigenvalues. Figure 3 (bottom) reports the quotient values for the Top-1 discrepancy direction as well as the averages over the Top-3 and Top-5 directions.

**Contrastive embedding alignment using KODA-identified samples.** To examine whether the discovered discrepancy slices are useful beyond visualization, we use them for targeted embedding alignment. In the unimodal setting, KODA identifies slices that are weakly grouped by CLIP but strongly grouped by DINOv2 on FFHQ dataset. We fine-tune CLIP on these selected samples to align its local geometry with DINOv2, following a kernel-based embedding-alignment objective of (Gong et al., 2025a). Table 2 shows that the aligned CLIP substantially improves its agreement with the KODA-discovered grouping and approaches the DINOv2 geometry on these slices. We observe a similar trend in the multimodal setting. On MSCOCO, KODA identifies image–caption pairs for which BLIP forms a clearer joint structure than CLIP. The corresponding t-SNE visualization in Figure 4 further shows that the aligned embedding forms a geometry closer to the target embedding on the same selected slice.

**Multimodal discrepancy reflects cross-modal alignment differences.** We further examine whether the multimodal discrepancies found by KODA mainly arise from cross-modal alignment differences or from general inter-model mismatch. Comparing image-only and joint image–text KODA directions, we find that image-only embeddings pro-

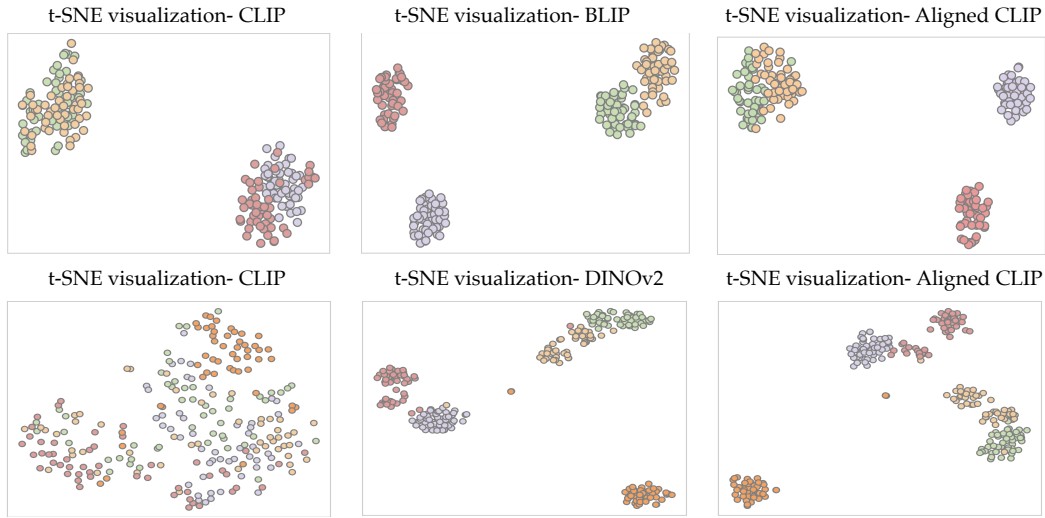

*Figure 4.* t-SNE visualization of KODA-selected samples before and after contrastive embedding alignment.

*Table 1.* Cross-modal retrieval on the full MSCOCO set and the KODA-selected subset. The KODA-selected subset amplifies the performance gap between CLIP and SigLIP.

| Model | Evaluation set | I2T R@1 | I2T R@5 | I2T R@10 | T2I R@1 | T2I R@5 | T2I R@10 | Avg. drop |
|-------|----------------|---------|---------|----------|---------|---------|----------|-----------|
| CLIP | Full | 32.64 | 57.88 | 68.10 | 28.60 | 53.04 | 64.46 | 0.00 |
| CLIP | KODA-selected | 18.00 | 44.00 | 55.00 | 19.00 | 46.00 | 56.00 | 11.12 |
| SigLIP | Full | 42.64 | 68.32 | 77.98 | 41.86 | 66.42 | 76.22 | 0.00 |
| SigLIP | KODA-selected | 34.00 | 68.00 | 78.00 | 35.00 | 62.00 | 73.00 | 3.91 |

*Table 2.* Contrastive embedding alignment on KODA-selected slices. AMI/NMI/ARI measure agreement between the embedding-induced clusters and the KODA-discovered grouping.

| Target | Model | AMI | NMI | ARI |
|--------|-------|-----|-----|-----|
| DINOv2 | CLIP | 0.25±.002 | 0.26±.002 | 0.19±.001 |
| | Aligned | 0.78±.004 | 0.78±.004 | 0.70±.012 |
| | Target | 0.83±.006 | 0.84±.006 | 0.77±.017 |
| BLIP | CLIP | 0.52±.003 | 0.53±.009 | 0.37±.004 |
| | Aligned | 0.91±.006 | 0.91±.008 | 0.90±.003 |
| | Target | 0.96±.008 | 0.96±.005 | 0.96±.006 |

duce noisier directions with weaker semantic separation, while joint image-text embeddings yield more coherent and concentrated clusters. Visual comparisons are in Figures 10 and 11. We also evaluate image-to-text and text-to-image retrieval on MSCOCO, comparing the KODA-selected subset with the full dataset. As in Table 1, the KODA-selected slice amplifies the retrieval gap between SigLIP and CLIP.

**Ablation Study.** To examine the sensitivity of KODA to major design parameters, we conduct ablation studies on the number of random Fourier features, the reference sample size, and the kernel function, discussed in Appendix C.5.

## 7. Conclusion and Limitations

In this work, we introduced *Contrastive Embedding Clustering* as a task for identifying sample-level structures that are organized differently across two embedding representations, and proposed KODA as a constrained kernel-based framework for solving this task. By formulating embedding comparison as a quadratic optimization problem with an explicit constraint on weak clusterability under a reference embedding, KODA directly localizes discrepancy directions that are strongly clustered in one embedding but diffuse or suppressed in the other, going beyond global kernel-difference spectral comparisons. We showed that the resulting non-convex problem admits efficient eigenvector-based characterization and developed scalable implementations through covariance-block reductions and random feature approximations. Across experiments, KODA identified fine-grained and interpretable contrastive clusters across multi-modal and uni-modal embeddings. As limitations, KODA requires a shared reference dataset and focuses on unsupervised discrepancy discovery through kernel-induced geometry; extending the framework to unmatched datasets, supervised or task-conditioned discrepancy notions, and stronger statistical guarantees for cluster interpretation are relevant directions for future work.

## Acknowledgments

This work is supported by a grant from the Research Grants Council of the Hong Kong Special Administrative Region, China, Project 14210725, and is also supported by CUHK Direct Research Grant with CUHK Project No. 4055164. The work is partially supported by a grant under 1+1+1 CUHK-CUHK(SZ)-GDSTC Joint Collaboration Fund. Also, the authors acknowledge the support from the Hong Kong Research Grants Council (RGC) and the Hong Kong PhD Fellowship Scheme (HKPFS) award supporting Youqi Wu's research. Finally, the authors sincerely thank the anonymous reviewers and meta-reviewer for their insightful suggestions and constructive feedback.

## Impact Statement

This work develops methods for comparing multi-modal embedding representations by identifying dataset-level discrepancy patterns between models. A positive impact is to support transparency and interpretability in the evaluation of widely used embedding interfaces, enabling more informed model selection, debugging, and analysis beyond aggregate benchmark metrics. As with other representation analysis tools, such methods could also be misused to exploit model-specific weaknesses or to support undesirable downstream applications if applied without appropriate safeguards. We therefore emphasize that KODA is intended for controlled evaluation and auditing purposes on shared reference datasets, and we encourage careful consideration of dataset provenance, privacy, and downstream use when applying embedding comparison techniques in practice.

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

# A. Additional Related Works

**Beyond downstream benchmarks: representation-level comparison.** Recent work has explored task-agnostic viewpoints on embedding comparison that deviate from pure downstream evaluation. Darrin et al. (2024) propose an information-theoretic approach to comparing embedding models based on notions of sufficiency/informativeness, enabling comparison without labeled tasks. The Platonic Representation Hypothesis (Huh et al., 2024) studies the extent to which independently trained models share representational structure, offering a complementary lens on when embeddings may share geometric organization.

**Kernel and spectral tools.** Kernel spectral methods provide classical machinery for studying grouping structure through eigensystems Chitta et al. (2012); Ghashami et al. (2016); Ullah et al. (2018); Sriperumbudur & Sterge (2022); Gedon et al. (2023). Kernel PCA (Schölkopf et al., 1998) and spectral clustering (Ng et al., 2001) relate eigenstructure of similarity matrices to latent clusters, while Random Fourier Features (Rahimi & Recht, 2007) enable scalable approximations for shift-invariant kernels. Building on these tools, spectral kernel-based embedding comparison methods have been proposed, including analyzing kernel differences constructed from a shared reference dataset (Jalali et al., 2025a). Our work follows this kernel-based, reference-set comparison paradigm but adopts an optimization-based formulation that explicitly enforces weak grouping under one embedding while maximizing grouping strength under another.

# B. Proofs

### B.1. Proof of Proposition 4.1

Consider Lagrangian multipliers $\lambda \geq 0$ for $x^\top K_B\, x \leq \epsilon$ and $\nu \in \mathbb{R}$ for $\|x\|_2^2 = 1$, and consider

$$\mathcal{L}(x, \lambda, \nu) = x^\top K_A\, x - \lambda(x^\top K_B\, x - \epsilon) - \nu(x^\top x - 1).$$

At a KKT point $(x^\star, \lambda^\star, \nu^\star)$, stationarity gives $2(K_A - \lambda^\star K_B - \nu^\star I)x^\star = 0$, yielding (9); complementary slackness yields (10). The strict feasibility assumption is a standard constraint qualification for the inequality constraint on the sphere.

### B.2. Proof of Proposition 4.2

First, note that $U_{t-1}^\top x = 0$ is equivalent to $x = P_{t-1}x$, which directly follows from the definition of $P_{t-1}$. For such $x$ and any symmetric $D$, we have $x^\top D x = x^\top (P_{t-1} D P_{t-1})x$ since $P_{t-1} = P_{t-1}^\top = P_{t-1}^2$. Applying this to $D = K_A, K_B$ yields the equivalence.

### B.3. Proof of Proposition 4.3

First note that

$$K_A - \lambda K_B = \Phi_A \Phi_A^\top - \lambda \Phi_B \Phi_B^\top = \Phi S_\lambda \Phi^\top.$$

For any $u \in \mathbb{R}^{d_1 + d_2}$,

$$(K_A - \lambda K_B)(\Phi u) = \Phi S_\lambda \Phi^\top \Phi u = \Phi S_\lambda (\Phi^\top \Phi) u = \Phi M_\lambda u.$$

Hence, if $M_\lambda u_\lambda = \eta_\lambda u_\lambda$, then with $x_\lambda = \Phi u_\lambda$ we have

$$(K_A - \lambda K_B)x_\lambda = \eta_\lambda x_\lambda,$$

thus $x_\lambda$ is an eigenvector of $K_A - \lambda K_B$ with eigenvalue $\eta_\lambda$. Moreover, $\eta_\lambda \neq 0$ implies $x_\lambda \neq 0$: if $\Phi u_\lambda = 0$ then $G u_\lambda = \Phi^\top \Phi u_\lambda = 0$ and thus $M_\lambda u_\lambda = S_\lambda G u_\lambda = 0$, forcing $\eta_\lambda = 0$.

It remains to show that $\eta_\lambda$ is also an eigenvalue of $M_\lambda$. This follows from the fact that for every matrix pair $A \in \mathbb{R}^{m \times n}$ and $B \in \mathbb{R}^{n \times m}$ (for any integers $m, n$), $AB$ and $BA$ share the same non-zero eigenvalues (including multiplicities). Applying this to the case $A = \Phi$ and $B = S_\lambda \Phi^\top$ shows that $\Phi S_\lambda \Phi^\top = K_A - \lambda K_B$ and $S_\lambda \Phi^\top \Phi = M_\lambda$ share the same non-zero eigenvalues. Since $K_A - \lambda K_B$ is symmetric, its eigenvalues are real, and particularly its largest eigenvalue $\eta_\lambda = \lambda_{\max}(K_A - \lambda K_B)$ is among the eigenvalues of $M_\lambda$. Therefore, $u_\lambda$ can be chosen as an eigenvector of $M_\lambda$ associated with $\eta_\lambda$, and the lifted vector $x_\lambda = \Phi u_\lambda$ is a principal eigenvector of $K_A - \lambda K_B$.

## B.4. Proof of Theorem 4.4

Let $D := d_1 + d_2$. Define $z(X) := [\phi_1(X); \phi_2(X)] \in \mathbb{R}^D$ and assume $\|z(X)\|_2^2 = \|\phi_1(X)\|_2^2 + \|\phi_2(X)\|_2^2 \le 2$ almost surely. Let

$$G := \mathbb{E}[z(X)z(X)^\top] \succeq 0, \qquad \widehat{G} := \frac{1}{n}\sum_{i=1}^n z(X_i)z(X_i)^\top \succeq 0,$$

and for $\lambda \ge 0$ define $S_\lambda := \mathrm{diag}(I_{d_1}, -\lambda I_{d_2})$,

$$B_\lambda := G^{1/2} S_\lambda G^{1/2}, \qquad \widehat{B}_\lambda := \widehat{G}^{1/2} S_\lambda \widehat{G}^{1/2}.$$

Note $B_\lambda$ and $\widehat{B}_\lambda$ are symmetric.

First, we prove a Frobenius concentration result for $\widehat{G}$ using the Hilbert-space Hoeffding's inequality. Let $W_i := z(X_i)z(X_i)^\top$ and $Z_i := W_i - G$, and therefore we have $\mathbb{E}[Z_i] = 0$ and $\widehat{G} - G = \frac{1}{n}\sum_{i=1}^n Z_i$. We bound $\|Z_i\|_F$ almost surely.

First, for any vector $a$, $\|aa^\top\|_F = \|a\|_2^2$: indeed, $\|aa^\top\|_F^2 = \mathrm{Tr}((aa^\top)^\top(aa^\top)) = \mathrm{Tr}(aa^\top aa^\top) = \|a\|_2^4$. Thus,

$$\|W_i\|_F = \|z(X_i)z(X_i)^\top\|_F = \|z(X_i)\|_2^2 \le 2 \quad \text{a.s.}$$

Also, by Jensen's inequality and the triangle inequality for the Frobenius norm,

$$\|G\|_F = \|\mathbb{E}[W_i]\|_F \le \mathbb{E}\|W_i\|_F \le 2.$$

Therefore,

$$\|Z_i\|_F = \|W_i - G\|_F \le \|W_i\|_F + \|G\|_F \le 4 \quad \text{a.s.}$$

We now apply the Hoeffding-type inequality for random vectors in Hilbert spaces (Sutherland et al., 2018) to the i.i.d. Hilbert-space-valued variables $Z_i$ in the Hilbert space of $D \times D$ matrices equipped with Frobenius norm. With $L = 4$, we obtain: for any $\delta \in (0, 1)$, with probability at least $1 - \delta$,

$$\|\widehat{G} - G\|_F = \Big\|\frac{1}{n}\sum_{i=1}^n Z_i\Big\|_F \le \frac{4}{\sqrt{n}}\Big(1 + \sqrt{2\log\tfrac{1}{\delta}}\Big) =: \eta_n(\delta). \tag{26}$$

Subsequently, we apply the Powers–Størmer inequality showing that for PSD matrices $A, B \succeq 0$ we have

$$\|A^{1/2} - B^{1/2}\|_F^2 \le \|A - B\|_*, \tag{27}$$

where $\|\cdot\|_*$ is the nuclear (trace) norm. Applying (27) to $A = \widehat{G}$ and $B = G$ yields

$$\|\widehat{G}^{1/2} - G^{1/2}\|_F \le \|\widehat{G} - G\|_*^{1/2}.$$

For any $D \times D$ matrix $X$, $\|X\|_* \le \sqrt{\mathrm{rank}(X)}\,\|X\|_F \le \sqrt{D}\,\|X\|_F$. Therefore,

$$\|\widehat{G}^{1/2} - G^{1/2}\|_F \le D^{1/4}\,\|\widehat{G} - G\|_F^{1/2}. \tag{28}$$

Then, we bound the norm difference $\|\widehat{B}_\lambda - B_\lambda\|_F$ using $\|G\|_2 \le 2$. To do so, we expand

$$\widehat{B}_\lambda - B_\lambda = (\widehat{G}^{1/2} - G^{1/2})S_\lambda \widehat{G}^{1/2} + G^{1/2} S_\lambda(\widehat{G}^{1/2} - G^{1/2}).$$

Taking the Frobenius norm and using submultiplicativity inequalities $\|AX\|_F \le \|A\|_2\|X\|_F$ and $\|XB\|_F \le \|B\|_2\|X\|_F$ result in the following inequality:

$$\|\widehat{B}_\lambda - B_\lambda\|_F \le \|S_\lambda\|_2\big(\|\widehat{G}^{1/2}\|_2 + \|G^{1/2}\|_2\big)\|\widehat{G}^{1/2} - G^{1/2}\|_F. \tag{29}$$

We now show that $\|\widehat{G}^{1/2}\|_2 \le \sqrt{2}$ and $\|G^{1/2}\|_2 \le \sqrt{2}$.

Since $G \succeq 0$, $\|G\|_2 = \lambda_{\max}(G) \le \mathrm{Tr}(G)$. Moreover,

$$\mathrm{Tr}(G) = \mathbb{E}\,\mathrm{Tr}(z(X)z(X)^\top) = \mathbb{E}\|z(X)\|_2^2 \le 2$$

As a result, $\|G\|_2 \le 2$ and thus $\|G^{1/2}\|_2 = \sqrt{\|G\|_2} \le \sqrt{2}$. Similarly, we have $\widehat{G} \succeq 0$ and $\|\widehat{G}\|_2 \le \mathrm{Tr}(\widehat{G})$. However, note that

$$\mathrm{Tr}(\widehat{G}) = \frac{1}{n}\sum_{i=1}^{n}\mathrm{Tr}(z(X_i)z(X_i)^\top) = \frac{1}{n}\sum_{i=1}^{n}\|z(X_i)\|_2^2 \le 2$$

which holds deterministically because each $\|z(X_i)\|_2^2 \le 2$. Therefore, $\|\widehat{G}\|_2 \le 2$ and $\|\widehat{G}^{1/2}\|_2 \le \sqrt{2}$. Consequently, the following holds

$$\|\widehat{G}^{1/2}\|_2 + \|G^{1/2}\|_2 \le 2\sqrt{2}. \tag{30}$$

Then, we substitute (28) and (30) into (29):

$$\|\widehat{B}_\lambda - B_\lambda\|_F \le 2\sqrt{2}\,\|S_\lambda\|_2\,D^{1/4}\,\|\widehat{G} - G\|_F^{1/2}.$$

On the event (26), we have $\|\widehat{G} - G\|_F \le \eta_n(\delta)$ and hence

$$\|\widehat{B}_\lambda - B_\lambda\|_F \le 2\sqrt{2}\,\|S_\lambda\|_2\,D^{1/4}\,\eta_n(\delta)^{1/2}.$$

Plugging in $\eta_n(\delta)$ from (26) leads to

$$\|\widehat{B}_\lambda - B_\lambda\|_F \le 2\sqrt{2}\,\|S_\lambda\|_2\,D^{1/4}\Big(\frac{4}{\sqrt{n}}\Big(1 + \sqrt{2\log\tfrac{1}{\delta}}\Big)\Big)^{1/2} = 8\sqrt{2}\,\|S_\lambda\|_2\,D^{1/4}\,n^{-1/4}\Big(1 + \sqrt{2\log\tfrac{1}{\delta}}\Big)^{1/2}.$$

Finally, note that the norm inequality $\|\cdot\|_2 \le \|\cdot\|_F$ implies that

$$\|\widehat{B}_\lambda - B_\lambda\|_2 \le \|\widehat{B}_\lambda - B_\lambda\|_F \le 8\sqrt{2}\,\|S_\lambda\|_2\,D^{1/4}\,n^{-1/4}\Big(1 + \sqrt{2\log\tfrac{1}{\delta}}\Big)^{1/2}.$$

Knowing that $8\sqrt{2} < 12$, the proof for the matrix concentration is complete. Finally, because $B_\lambda$ and $\widehat{B}_\lambda$ are symmetric matrices, the rank-one Davis–Kahan $\sin\Theta$ bound applies as follows: Given that $\gamma_\lambda := \lambda_1(B_\lambda) - \lambda_2(B_\lambda) > 0$ and $v_1, \widehat{v}_1$ are unit-norm top eigenvectors, then the following inequality holds

$$\sin\angle(\widehat{v}_1, v_1) \le \frac{\|\widehat{B}_\lambda - B_\lambda\|_2}{\gamma_\lambda}.$$

The proof is hence complete.

### B.5. Proof of Theorem 5.1

We first prove the Frobenius concentration bound (24) by applying a Hilbert-space Hoeffding inequality directly to the random matrices (viewed as vectors under the Frobenius norm). We then derive the eigenspace bound (25) via Davis–Kahan.

Note that, by assumption, $\widetilde{K} = \frac{1}{r}\sum_{\ell=1}^{r} K^{(\ell)}$, where the matrices $K^{(\ell)}$ are i.i.d., $\mathbb{E}[K^{(\ell)}] = K$, and $|K_{ij}^{(\ell)}| \le 1$ almost surely for all $i, j$. Define the normalized matrices

$$A^{(\ell)} := \frac{1}{n}K^{(\ell)}, \qquad A := \frac{1}{n}K, \qquad \widetilde{A} := \frac{1}{n}\widetilde{K} = \frac{1}{r}\sum_{\ell=1}^{r} A^{(\ell)}.$$

Then $\mathbb{E}[A^{(\ell)}] = A$, and

$$\widetilde{A} - A = \frac{1}{r}\sum_{\ell=1}^{r}\big(A^{(\ell)} - A\big).$$

We work in the Hilbert space $(\mathbb{R}^{n \times n}, \langle \cdot, \cdot \rangle_F)$ where $\|M\| = \|M\|_F$. We let $X_\ell := A^{(\ell)} - A$. Then, $\{X_\ell\}_{\ell=1}^r$ are i.i.d. random elements of this Hilbert space with $\mathbb{E}[X_\ell] = 0$. Next, we derive applicable upper-bounds on $\|X_\ell\|_F$ that hold with provable probability. Since $|K_{ij}^{(\ell)}| \leq 1$ holds deterministically, we have the following for every index $\ell$,

$$\|A^{(\ell)}\|_F^2 = \sum_{i=1}^n \sum_{j=1}^n \left(\frac{K_{ij}^{(\ell)}}{n}\right)^2 \leq \sum_{i=1}^n \sum_{j=1}^n \frac{1}{n^2} = 1,$$

and hence $\|A^{(\ell)}\|_F \leq 1$ almost surely. Similarly, using $|K_{ij}| \leq 1$ for all $i, j$,

$$\|A\|_F^2 = \sum_{i=1}^n \sum_{j=1}^n \left(\frac{K_{ij}}{n}\right)^2 \leq \sum_{i=1}^n \sum_{j=1}^n \frac{1}{n^2} = 1,$$

Therefore, $\|A\|_F \leq 1$ holds. Hence, using the triangle inequality, we can show

$$\|X_\ell\|_F = \|A^{(\ell)} - A\|_F \leq \|A^{(\ell)}\|_F + \|A\|_F \leq 2$$

Thus, the centered summands are uniformly bounded in norm by $L = 2$.

We now apply the Hoeffding inequality for random vectors (Sutherland et al., 2018) to the i.i.d. sequence $\{X_\ell\}_{\ell=1}^r$, with $\ell_2$-norm upper-bound $L = 2$. This shows that for every $\delta > 0$, the following holds with probability at least $1 - \delta$:

$$\left\|\frac{1}{r} \sum_{\ell=1}^r X_\ell\right\|_F \leq \frac{2}{\sqrt{r}}\left(1 + \sqrt{2 \log \frac{1}{\delta}}\right).$$

Recalling that $\frac{1}{r} \sum_{\ell=1}^r X_\ell = \widetilde{A} - A = \frac{1}{n}(\widetilde{K} - K)$, we obtain

$$\frac{1}{n}\|\widetilde{K} - K\|_F = \|\widetilde{A} - A\|_F \leq \frac{2}{\sqrt{r}}\left(1 + \sqrt{2 \log \frac{1}{\delta}}\right).$$

Multiplying both sides by $n$ yields the claimed Frobenius bound (24).

Subsequently, note that both $K$ and $\widetilde{K}$ are symmetric matrices. Let $U$ and $\widetilde{U}$ be the top-$q$ eigenspaces of $K$ and $\widetilde{K}$ (represented by $n \times q$ matrices with orthonormal columns), and assume the eigengap $\Delta_q(K) = \lambda_q(K) - \lambda_{q+1}(K) > 0$. A standard Davis–Kahan $\sin \Theta$ bound gives

$$\left\|\sin \Theta(\widetilde{U}, U)\right\|_F \leq \frac{\|\widetilde{K} - K\|_2}{\Delta_q(K)}.$$

Finally, for any matrix $E$, $\|E\|_2 \leq \|E\|_F$. Applying this with $E = \widetilde{K} - K$ yields

$$\left\|\sin \Theta(\widetilde{U}, U)\right\|_F \leq \frac{\|\widetilde{K} - K\|_F}{\Delta_q(K)}.$$

The above inequality completes the proof.

## C. Additional Numerical Results

### C.1. Experiment Details

All experiments are conducted in the covariance-operator formulation of KODA. To ensure numerical stability and efficiency, the associated generalized eigenvalue problems are solved via Cholesky decomposition of the constrained covariance operator, following standard practice in kernel-based spectral methods. Throughout all experiments, we adopt a Gaussian (RBF) kernel without other specifications. To enable scalable computation for both unimodal and multi-modal settings, kernel features are approximated using Random Fourier Features (RFF). Unless otherwise specified, we use 3,000 random Fourier features to approximate each Gaussian kernel. The kernel bandwidth $\sigma$ is selected following common practice in the representation comparison literature. We adopt the same bandwidth selection strategy as in prior work (Zhang et al.,

2024) (Jalali et al., 2025a). Specifically, for each pair of embeddings under comparison, we tune the kernel bandwidths such that the leading eigenvalues of the resulting kernel matrices are of comparable magnitude across models, ensuring that neither embedding dominates the optimization due to scale differences. For the quadratic constraint in KODA, the threshold $\epsilon$ controls the degree of weak clustering enforced under the constrained embedding. Unless otherwise stated, we set $\epsilon$ to the 0.5 quantile of the eigenvalues of the constrained model. All experiments are performed on two NVIDIA RTX 4090 GPUs.

## C.2. Sanity Check of Unimodal Comparison

**Visualizing Pairwise Discrepancies via Kernel Difference Heatmaps.** Figure 5 shows normalized RBF kernel similarities induced by CLIP and DINOv2 on ImageWoof (ImageNet-1k dog breeds), together with their difference matrix. The difference heatmap exhibits structured, high-magnitude regions, where darker values indicate stronger mismatch between the two embeddings under the same similarity metric. These mismatches concentrate on specific breed-level relationships, suggesting that the discrepancies arise along meaningful semantic directions rather than random fluctuations.

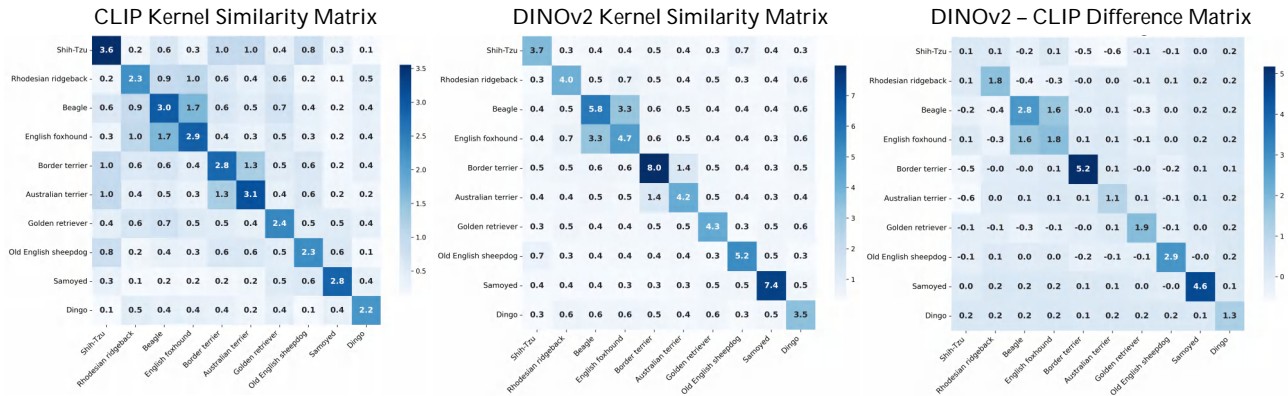

*Figure 5.* Kernel similarity heatmaps induced by CLIP and DINOv2 on the ImageNet-1k dog breeds, together with their difference. (scaled by 100 for better visualization.)

**Identified Discrepancy Directions Consistent with Kernel Difference Structures.** Based on the above normalized RBF kernel difference between the two embeddings, we identify the dog-breed categories associated with the largest aggregated pairwise mismatches as a ground-truth reference of semantic discrepancy. We then apply KODA to the same dataset without using any label information to discover discrepancy directions between the two embeddings. For each discovered direction, we select the top-6 images for visualization. As shown in Figure 6, the top discrepancy directions recovered by KODA correspond closely to the most mismatched dog-breed categories identified from the kernel difference matrix.

## C.3. Additional Results on Unimodal Comparison

We provide additional qualitative results for unimodal embedding comparison to complement the main experiments. In particular, we analyze the dominant discrepancy directions discovered by KODA between DINOv2 and CLIP embeddings on AFHQ (Choi et al., 2020) dataset. We consider both asymmetric comparison settings: (i) directions that are weakly clustered under CLIP while being strongly grouped under DINOv2, and (ii) directions that are weakly clustered under DINOv2 while being strongly grouped under CLIP. For each setting, we visualize the top discrepancy components obtained from KODA by inspecting the samples associated with the leading directions, as shown in the Figure 7 and Figure 8. Also, we compare directions identified by KODA with those from the SPEC (Jalali et al., 2025a) baseline, using the same settings. We quantify the strength of a discrepancy direction $x$ using the generalized Rayleigh quotient $\frac{x^\top K_1 x}{x^\top K_2 x}$, where $K_1$ and $K_2$ are normalized RBF kernel matrices induced by the two embeddings. Since $x^\top K x$ measures how strongly direction $x$ is expressed under kernel $K$, larger quotient values indicate stronger directional asymmetry, i.e., directions emphasized by $K_1$ but suppressed by $K_2$. Figure 9 reports the quotient values of KODA's Top-1 direction and the averages over Top-3 and Top-5 directions across different quantiles, together with SPEC's Top-1 direction. Across all constraint levels, KODA consistently achieves substantially larger quotient values; notably, at $q = 0.1$, KODA's Top-1 direction is about $20\times$ stronger than SPEC, and even the Top-5 average remains clearly above SPEC throughout. These additional results further demonstrate the ability of KODA to disentangle directional discrepancies that depend on the choice of reference embedding, even in unimodal settings.

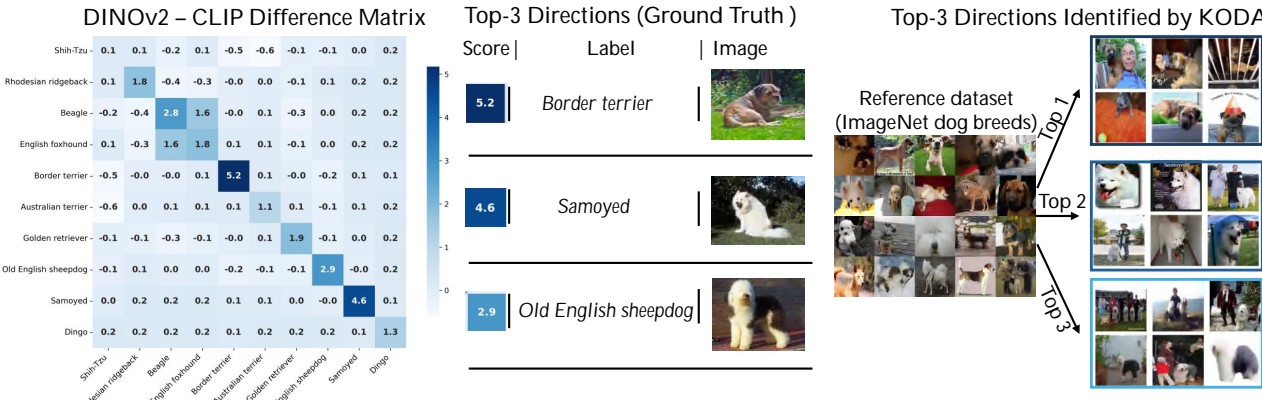

*Figure 6.* Consistency between dominant kernel mismatches (ground truth) and discrepancy directions identified by KODA on ImageNet dog breeds. **Left:** the kernel difference matrix between DINOv2 and CLIP computed using normalized RBF kernels. **Middle:** the top-3 ground-truth dog breeds associated with the largest aggregated mismatch scores in the difference matrix, together with representative images. **Right:** representative samples from the top-3 discrepancy directions identified by KODA without using label information.

# Top-10 mismatch directions discovered by KODA

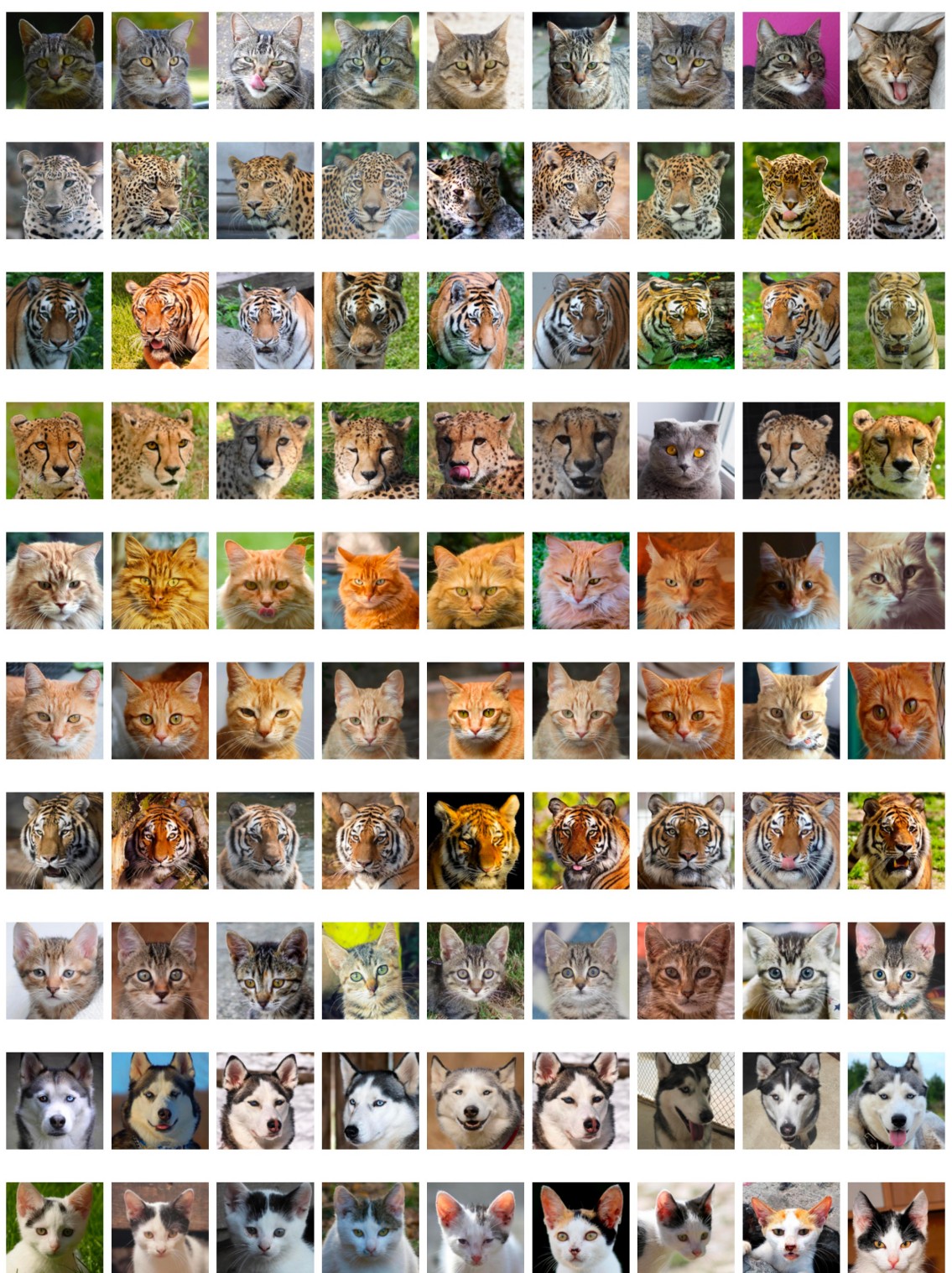

*Figure 7.* Top-10 DINOv2 dominant directions relative to CLIP on the AFHQ dataset identified by KODA, visualized via representative samples for each direction.

# Top-10 mismatch directions discovered by KODA

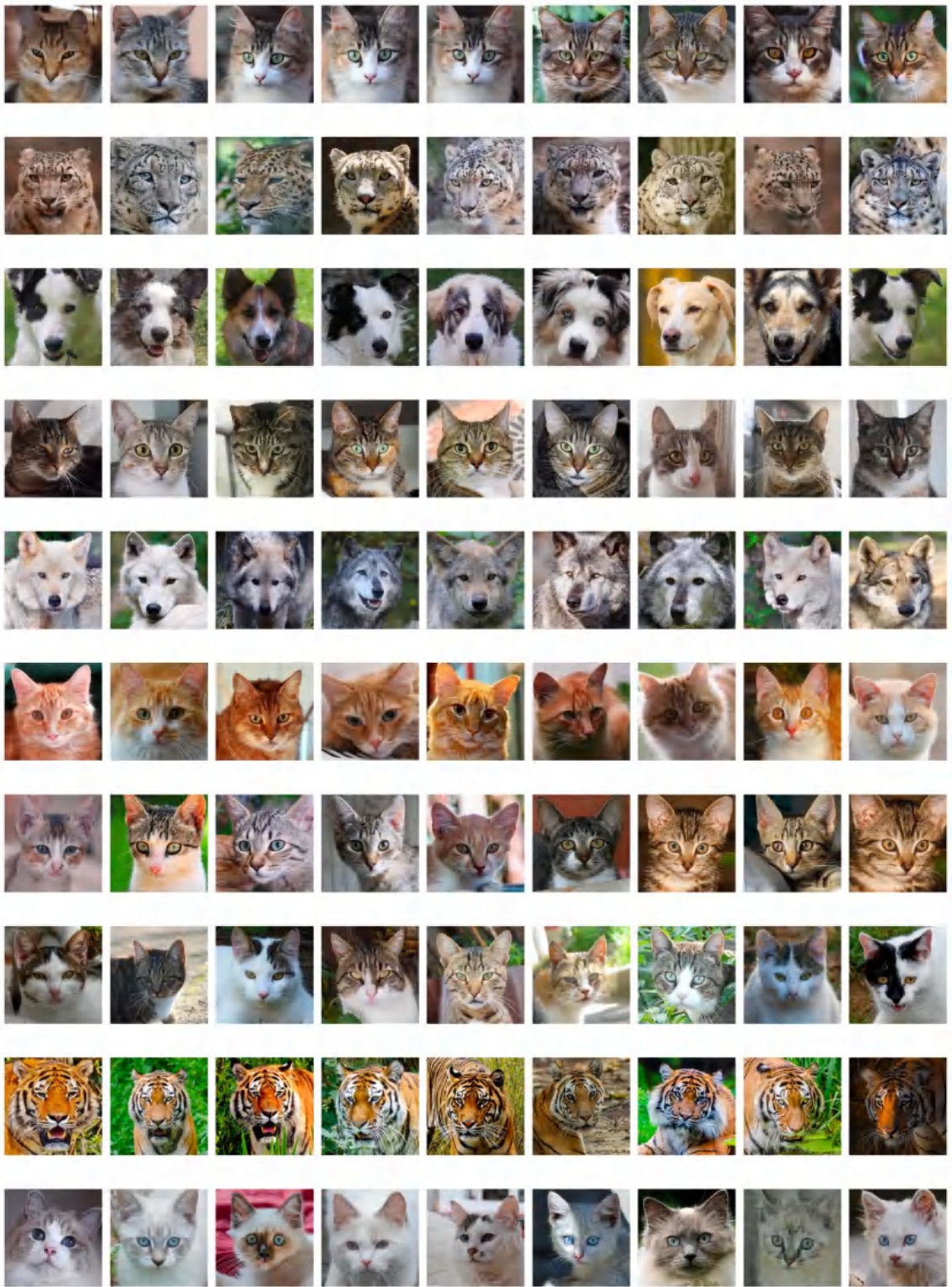

*Figure 8.* Top-10 CLIP dominant directions relative to DINOv2 on the AFHQ dataset identified by KODA, visualized via representative samples for each direction.

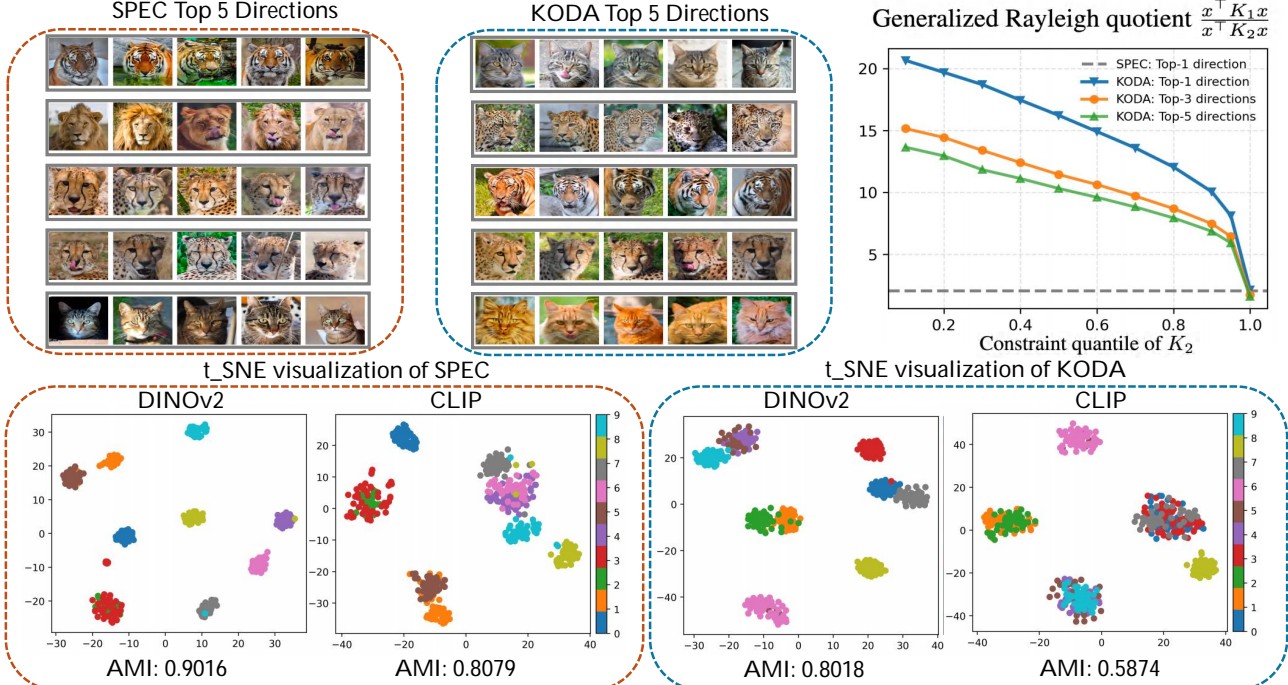

*Figure 9.* Left: Visualization of the top-5 mismatch directions of DINOv2 and CLIP on the AFHQ dataset discovered by KODA (ours) and SPEC (baseline), respectively. Right: Generalized Rayleigh quotient $\frac{x^{\top} K_1 x}{x^{\top} K_2 x}$ w.r.t. the constraint on $K_2$. (The quotient can be interpreted as a multiplicative measure of how strongly a given direction is represented in DINOv2 relative to CLIP.)

## C.4. Additional Results on Multimodal Comparison

We further provide additional results for multimodal embedding comparison on the MS-COCO dataset. In this setting, we analyze discrepancy directions discovered by KODA across a diverse set of vision–language models, including CLIP, OpenCLIP, BLIP, SigLIP, and SigLIP2. For each pair of multimodal embeddings, we construct joint image–text kernels and apply KODA to identify dominant discrepancy directions under asymmetric constraints. We visualize the samples associated with the top discrepancy components by inspecting the image–text pairs corresponding to the leading directions, as shown in the Figure 12-18.

These visualizations highlight how different multimodal models organize paired image–text data in distinct ways, even when trained on similar objectives or datasets. Across different model combinations, the dominant discrepancy directions correspond to different subsets of samples, reflecting variations in how visual and textual information is jointly encoded. These additional results complement the main experiments by illustrating the generality of KODA across a wide range of multimodal embedding families.

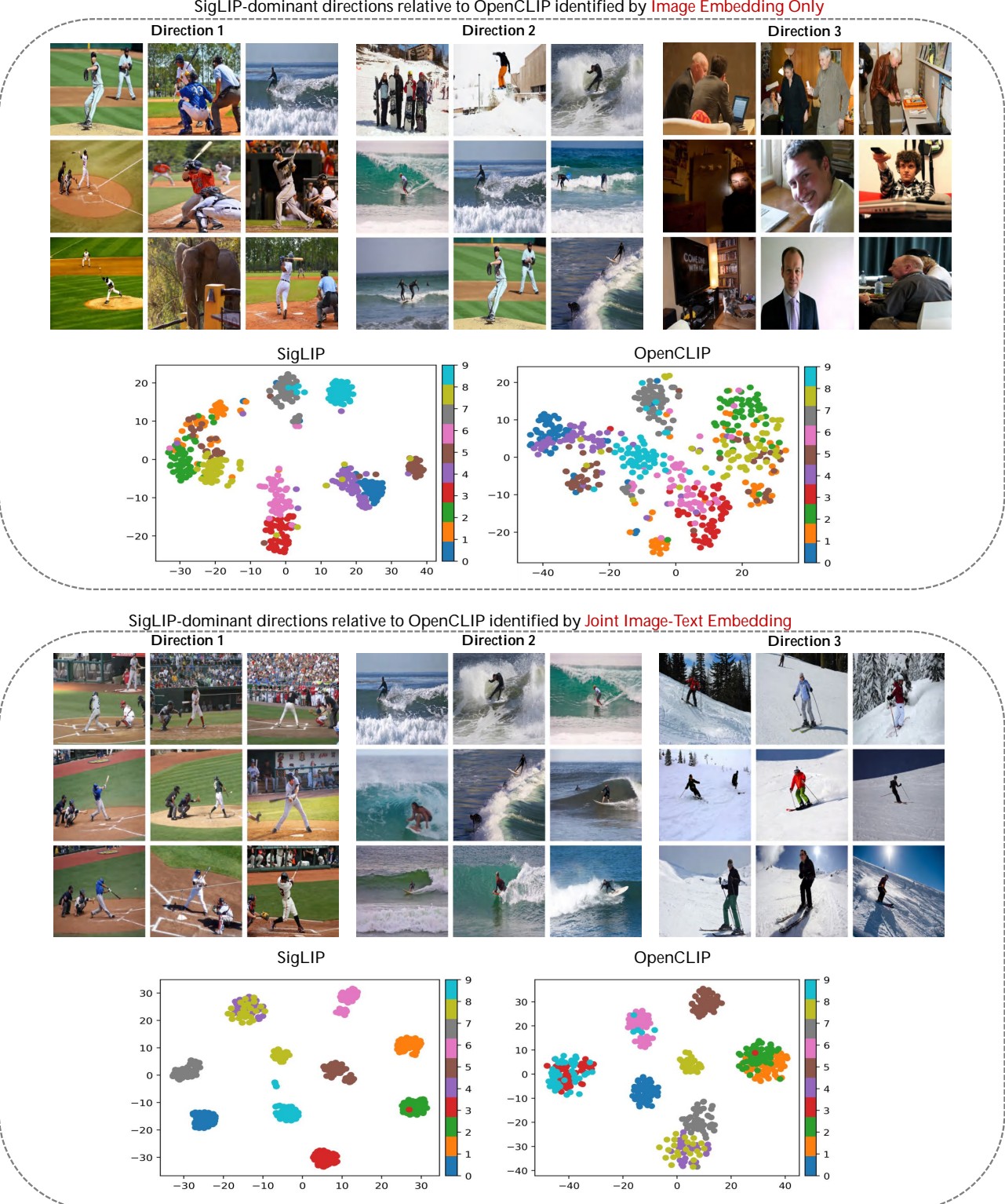

*Figure 10.* Multimodal discrepancy analysis of SigLIP dominant directions relative to OpenCLIP on the MSCOCO dataset.

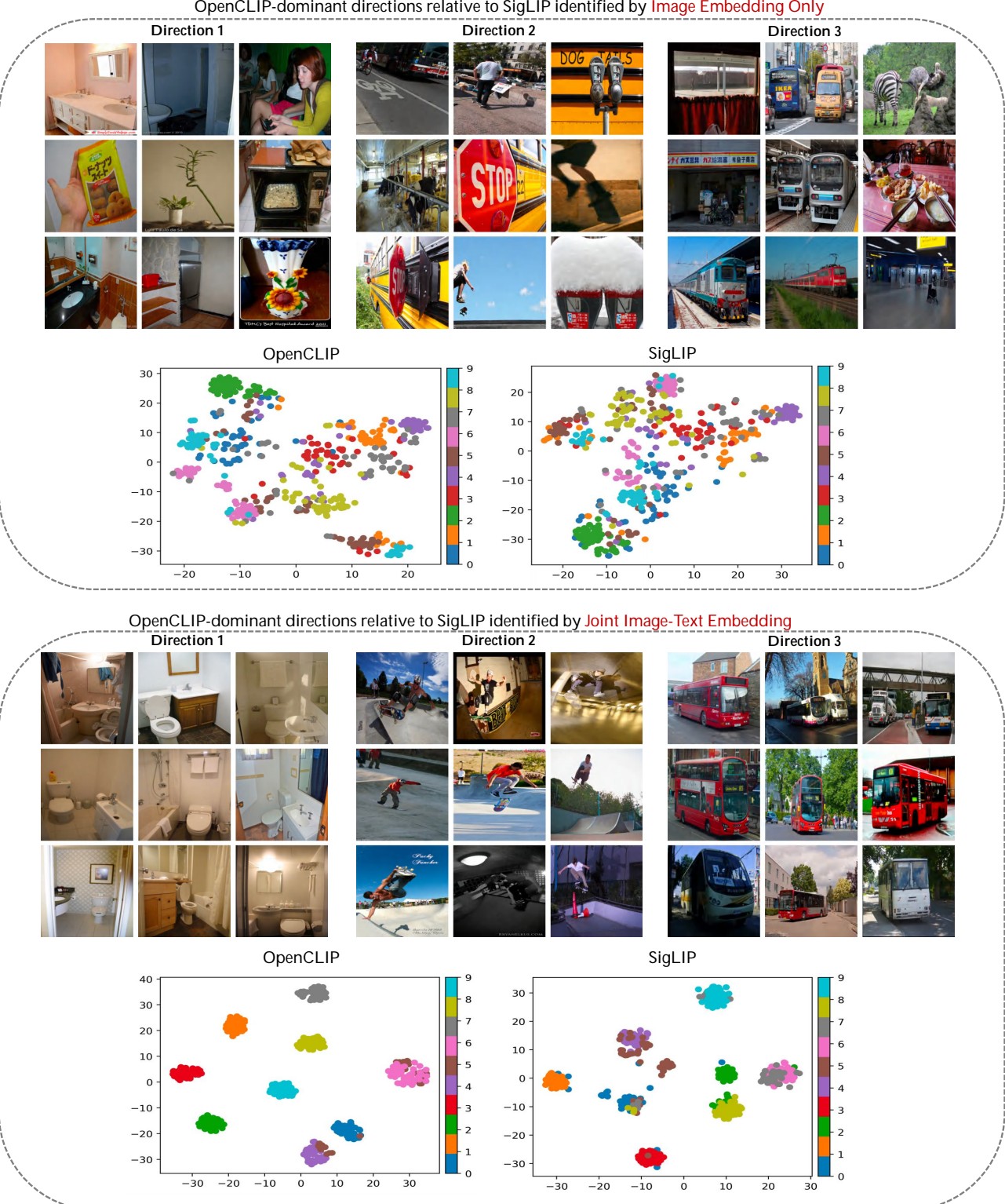

*Figure 11.* Multimodal discrepancy analysis of OpenCLIP dominant directions relative to SigLIP on the MSCOCO dataset.

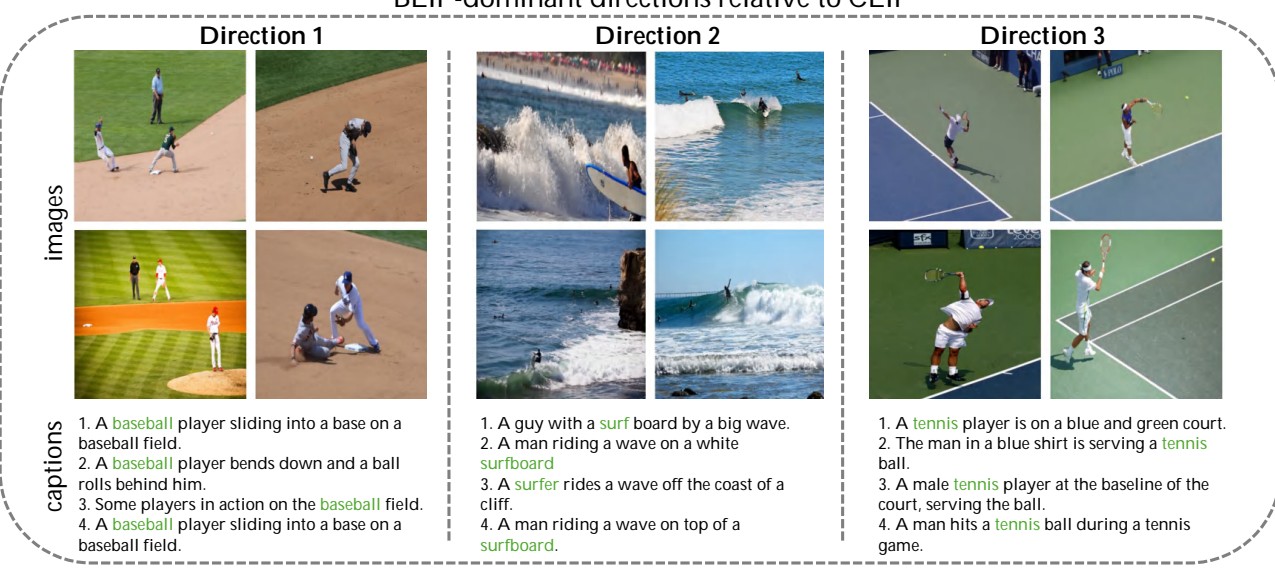

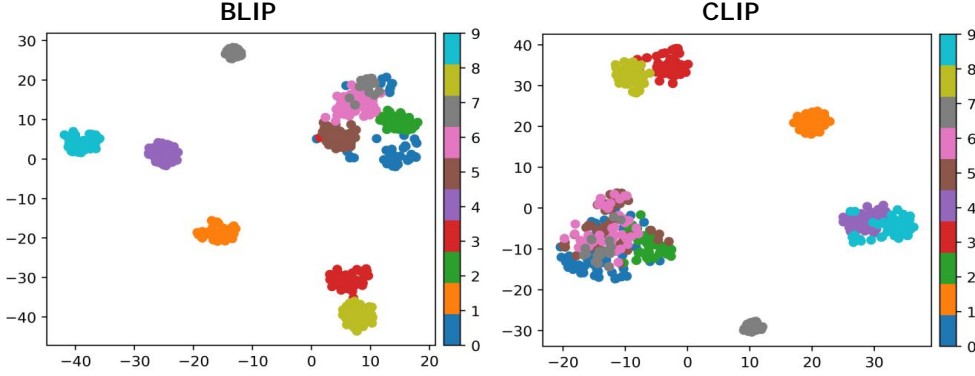

*Figure 12.* Multimodal discrepancy analysis of BLIP dominant directions relative to CLIP on the MSCOCO dataset. **Top:** Representative image–caption pairs corresponding to the Top-3 discrepancy directions identified by KODA. **Bottom:** t-SNE visualization of Top-10 discrepancy directions using BLIP and CLIP embeddings respectively.

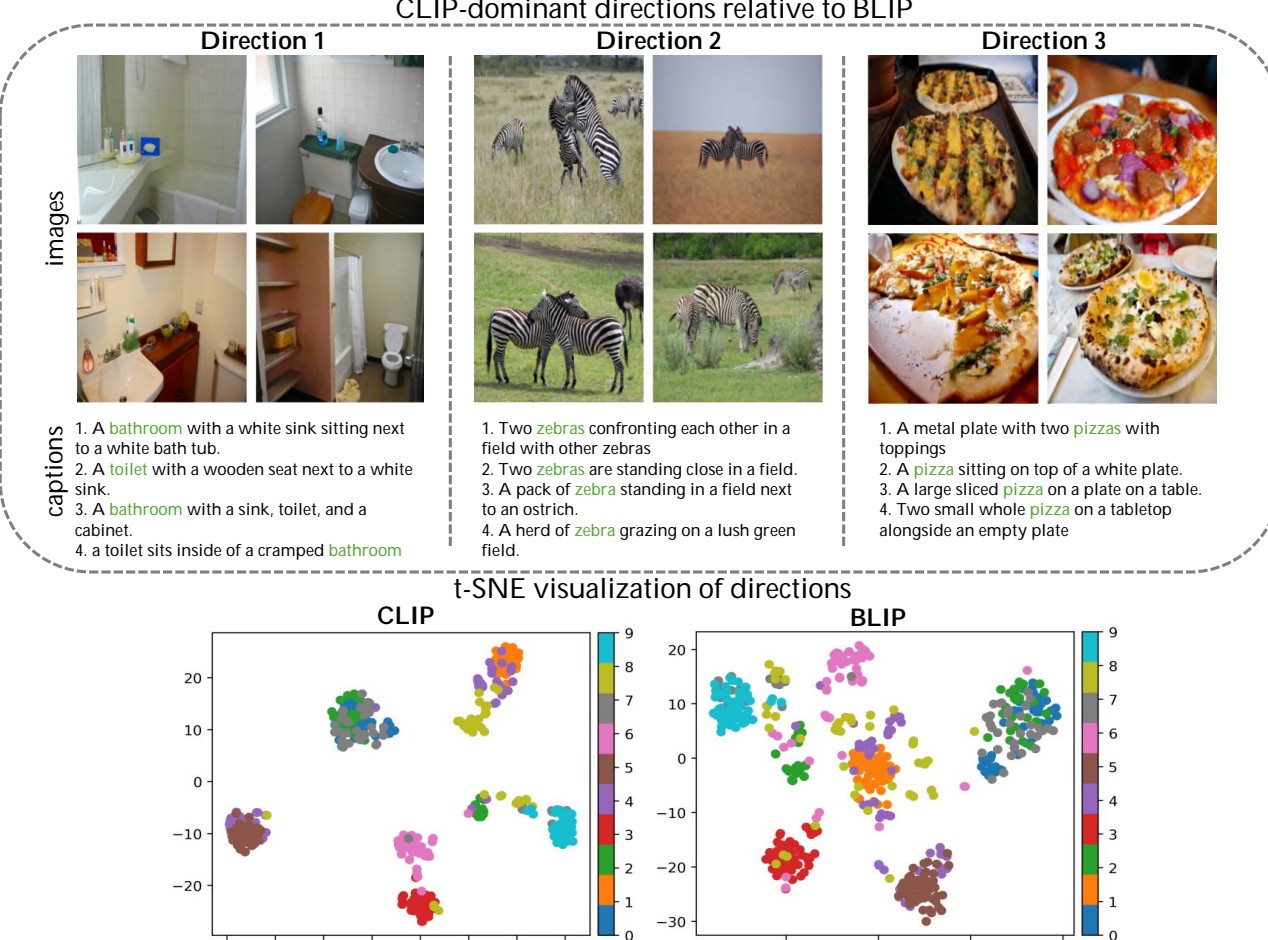

*Figure 13.* Multimodal discrepancy analysis of CLIP dominant directions relative to BLIP on the MSCOCO dataset. **Top:** Representative image–caption pairs corresponding to the Top-3 discrepancy directions identified by KODA. **Bottom:** t-SNE visualization of Top-10 discrepancy directions using CLIP and BLIP embeddings respectively.

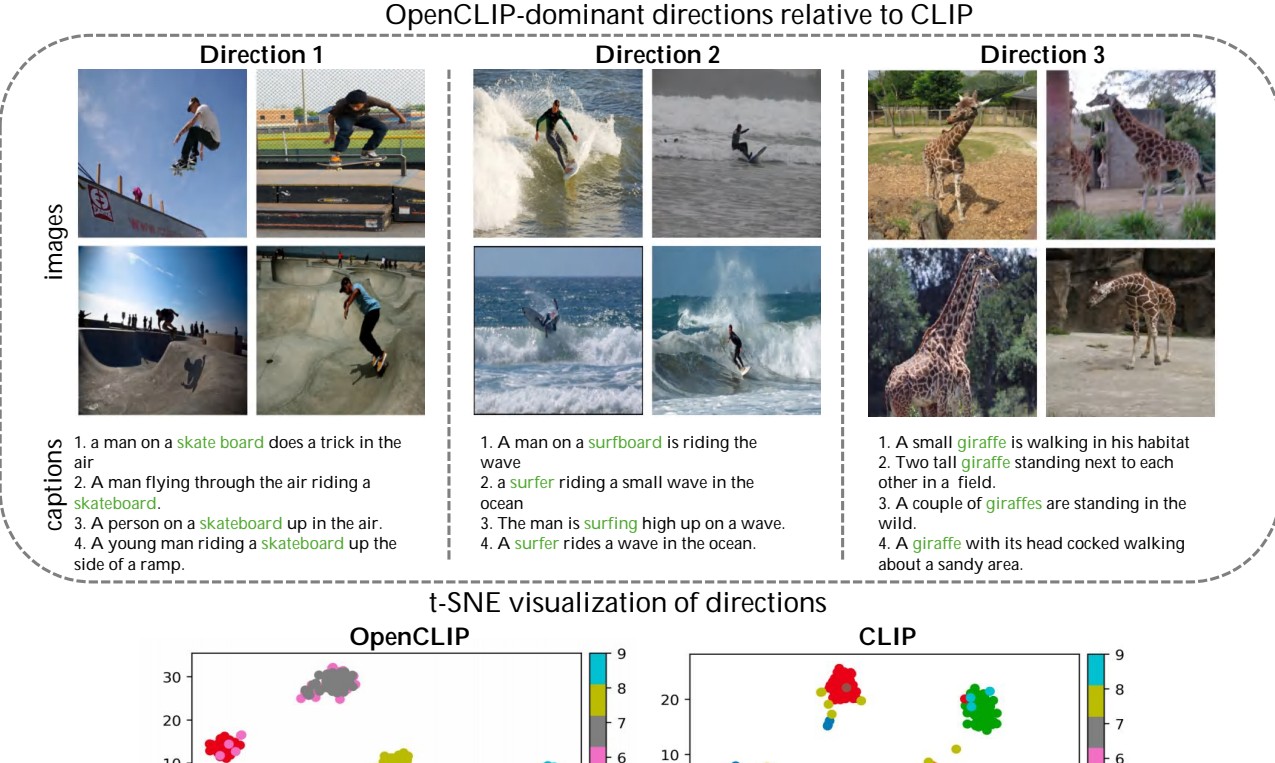

*Figure 14.* Multimodal discrepancy analysis of OpenCLIP dominant directions relative to CLIP on the MSCOCO dataset. **Top:** Representative image–caption pairs corresponding to the Top-3 discrepancy directions identified by KODA. **Bottom:** t-SNE visualization of Top-10 discrepancy directions using OpenCLIP and CLIP embeddings respectively.

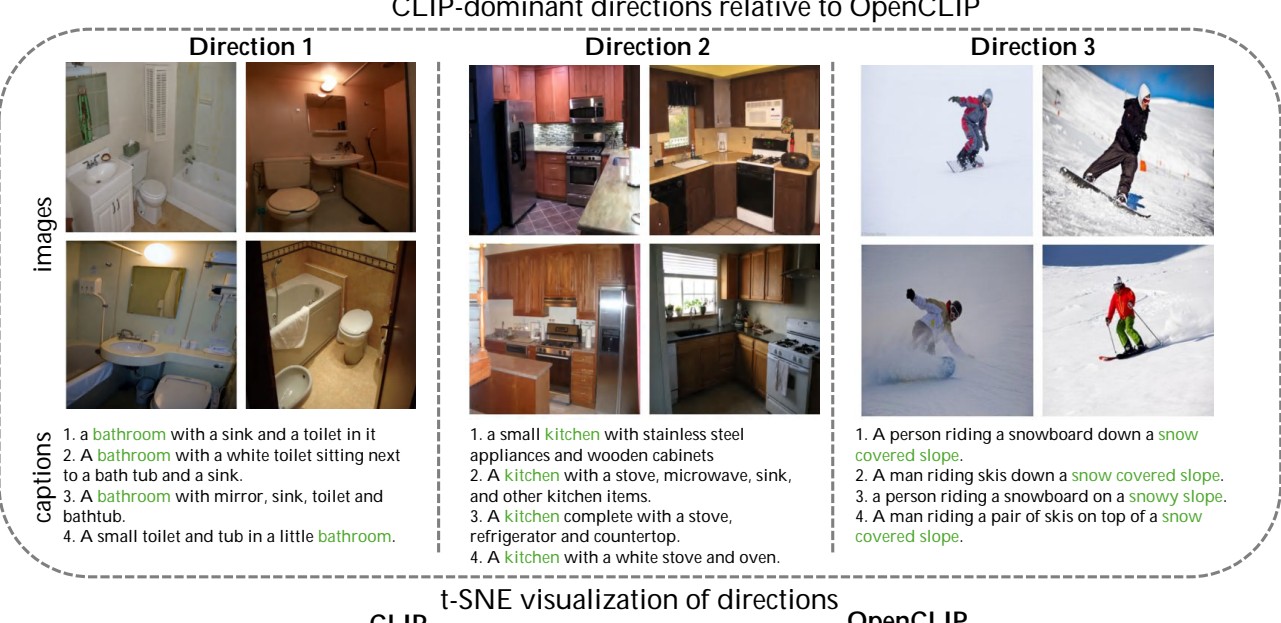

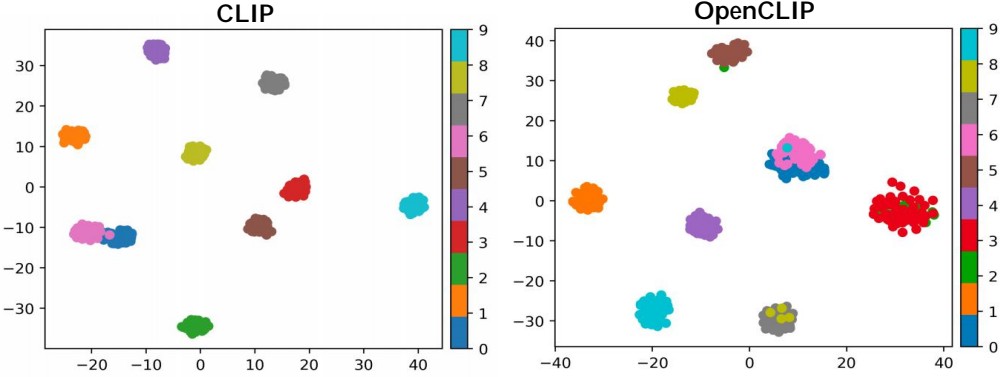

*Figure 15.* Multimodal discrepancy analysis of CLIP dominant directions relative to OpenCLIP on the MSCOCO dataset. **Top:** Representative image–caption pairs corresponding to the Top-3 discrepancy directions identified by KODA. **Bottom:** t-SNE visualization of Top-10 discrepancy directions using CLIP and OpenCLIP embeddings respectively.

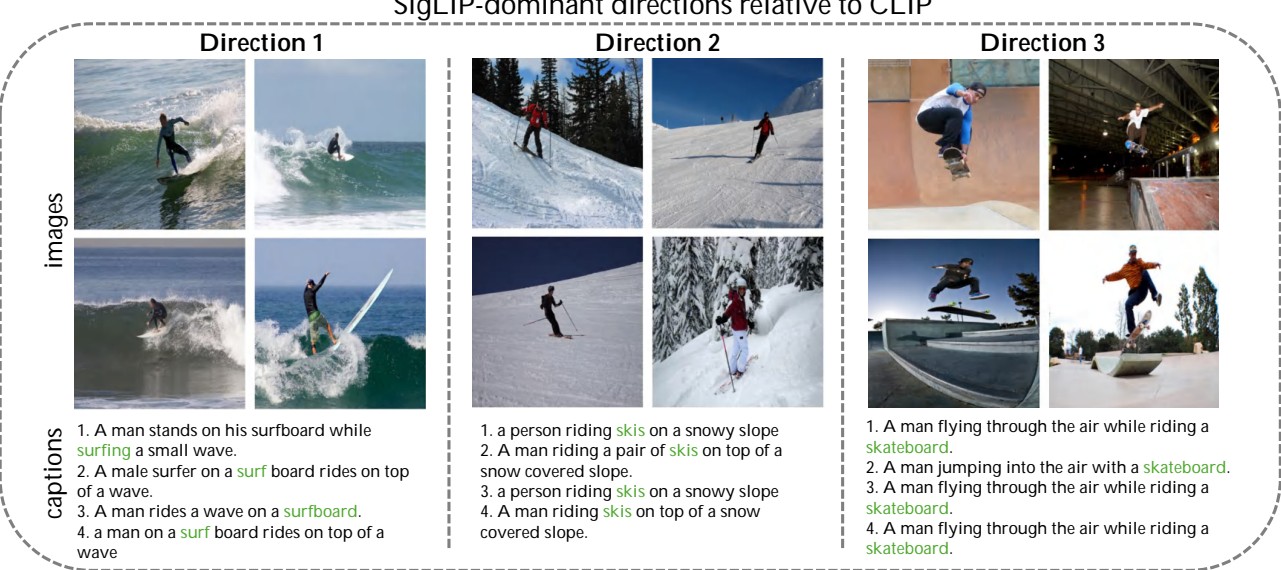

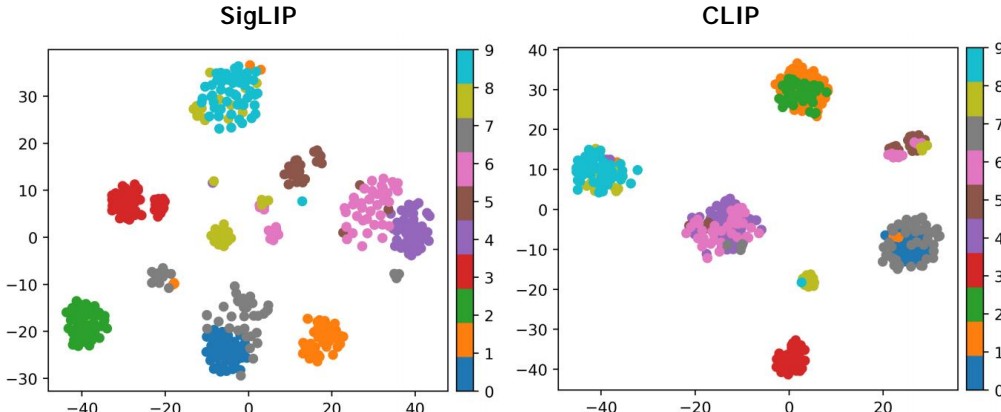

*Figure 16.* Multimodal discrepancy analysis of SigLIP dominant directions relative to CLIP on the MSCOCO dataset. **Top:** Representative image–caption pairs corresponding to the Top-3 discrepancy directions identified by KODA. **Bottom:** t-SNE visualization of Top-10 discrepancy directions using SigLIP and CLIP embeddings respectively.

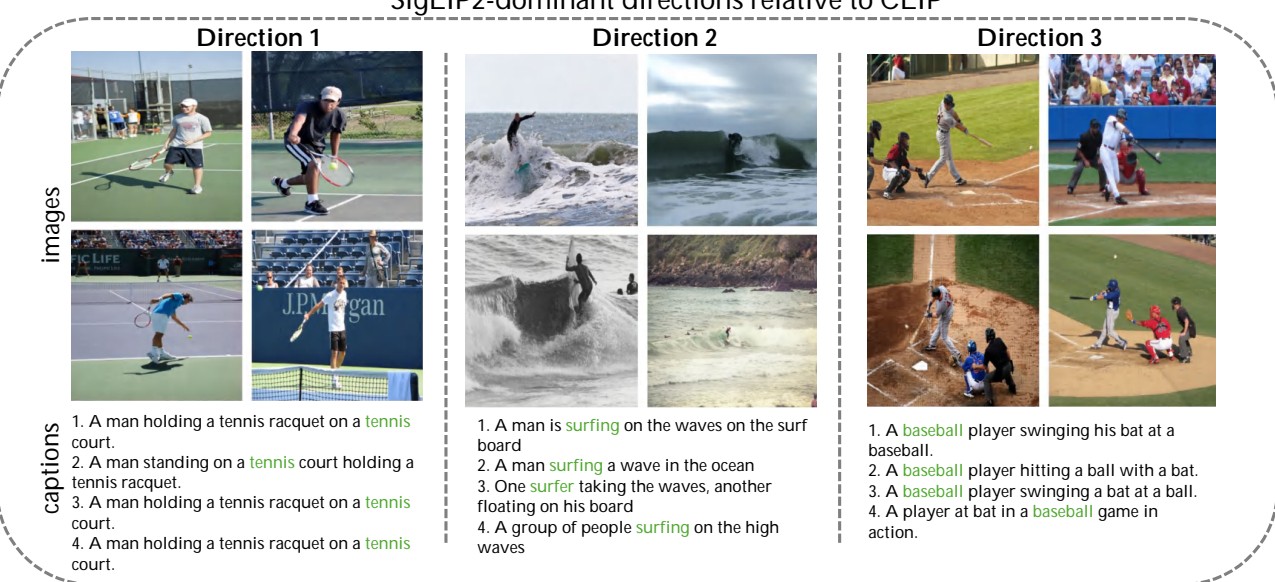

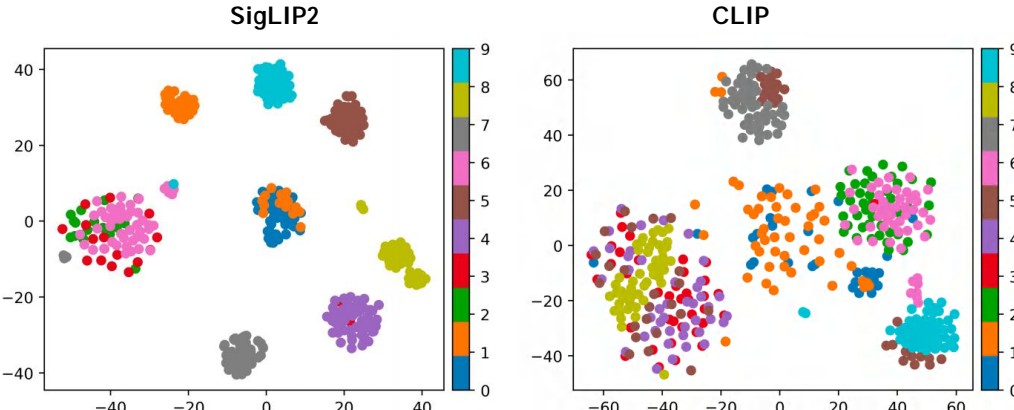

*Figure 17.* Multimodal discrepancy analysis of SigLIP2 dominant directions relative to CLIP on the MSCOCO dataset. **Top:** Representative image–caption pairs corresponding to the Top-3 discrepancy directions identified by KODA. **Bottom:** t-SNE visualization of Top-10 discrepancy directions using SigLIP2 and CLIP embeddings respectively.

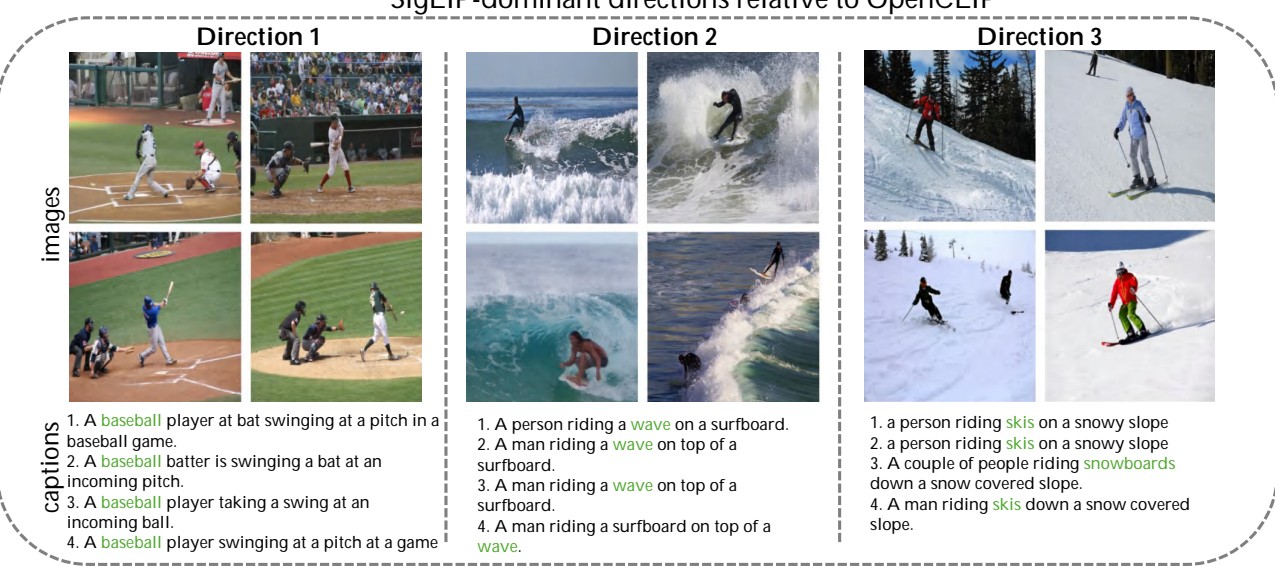

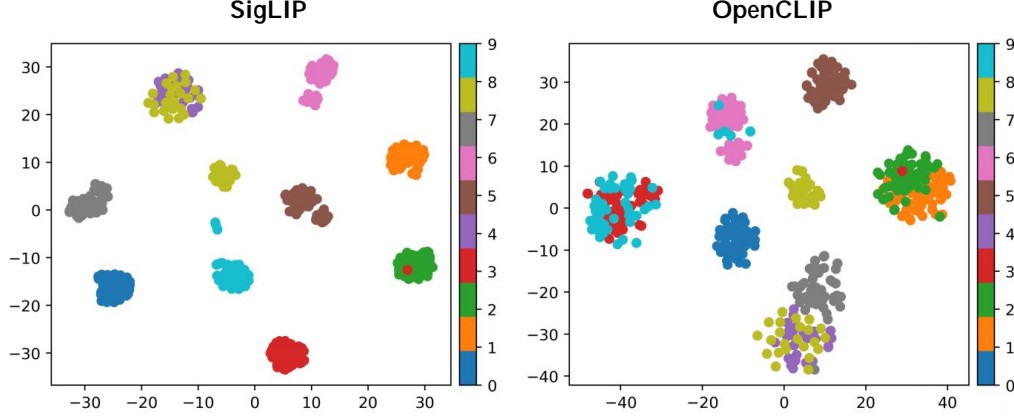

*Figure 18.* Multimodal discrepancy analysis of SigLIP dominant directions relative to OpenCLIP on the MSCOCO dataset. **Top:** Representative image–caption pairs corresponding to the Top-3 discrepancy directions identified by KODA. **Bottom:** t-SNE visualization of Top-10 discrepancy directions using SigLIP and OpenCLIP embeddings respectively.

## C.5. Ablation Study

We conduct a set of ablation studies to examine the sensitivity of KODA to key design choices, including the number of random Fourier features, the size of the reference sample set, and the choice of kernel function. All ablation experiments are conducted using the same experimental protocol as in the main results, with only the specified factor varied while keeping other settings fixed.

**Number of Random Fourier Features.** We examine the effect of the number of random Fourier features used to approximate the Gaussian kernel by varying the feature dimensionality $r \in \{500, 1000, 2000, 3000\}$. For each setting, we visualize the dominant discrepancy directions discovered by KODA in Figure 19. As the number of random features increases, the discovered discrepancy directions become progressively more coherent and visually well-separated. In particular, higher-dimensional approximations lead to cleaner and more stable grouping patterns, while lower-dimensional approximations exhibit increased noise in the dominant directions. These results indicate that sufficiently rich random feature approximations are beneficial for stable discrepancy discovery, and motivate our choice of $r = 3000$ in the main experiments.

**Kernel Function.** We investigate the effect of the kernel function by comparing Gaussian (RBF) kernels with cosine similarity kernels. For each kernel choice, we apply KODA using the same constraint setting and visualize the dominant discrepancy directions. As Figure 20 shows, different kernel functions lead to different discrepancy patterns, reflecting the distinct geometric properties emphasized by each kernel. In particular, Gaussian kernels capture local neighborhood structure based on Euclidean distance, whereas cosine similarity kernels emphasize angular relationships between representations. As a result, the dominant discrepancy directions discovered under different kernels correspond to different groupings of samples.

**Reference Sample Size.** We study the effect of the reference sample size by varying the number of samples $n \in \{2000, 4000, 8000, 16000\}$ while keeping all other settings fixed. For each choice of $n$, we apply KODA and visualize the dominant discrepancy directions in Figure 21. As the sample size increases, the discovered discrepancy directions become increasingly stable and consistent. In particular, when $n = 16,000$, the resulting discrepancy components exhibit highly stable grouping patterns, indicating that sufficient reference coverage is important for reliable discrepancy discovery. At smaller sample sizes, the overall structure of the discrepancy directions is preserved, albeit with increased variability in the visualizations. Notably, performing spectral decomposition directly on kernel matrices of size exceeding $10,000 \times 10,000$ is often impractical on modern GPUs due to memory and computational constraints. By operating in the covariance space induced by random feature representations, KODA reduces the effective dimensionality of the spectral problem to the feature dimension (e.g., 6,000 in our experiments), making eigen-decomposition feasible even when the number of reference samples is large. This formulation enables stable discrepancy analysis at larger sample sizes without requiring explicit construction or decomposition of full kernel matrices.

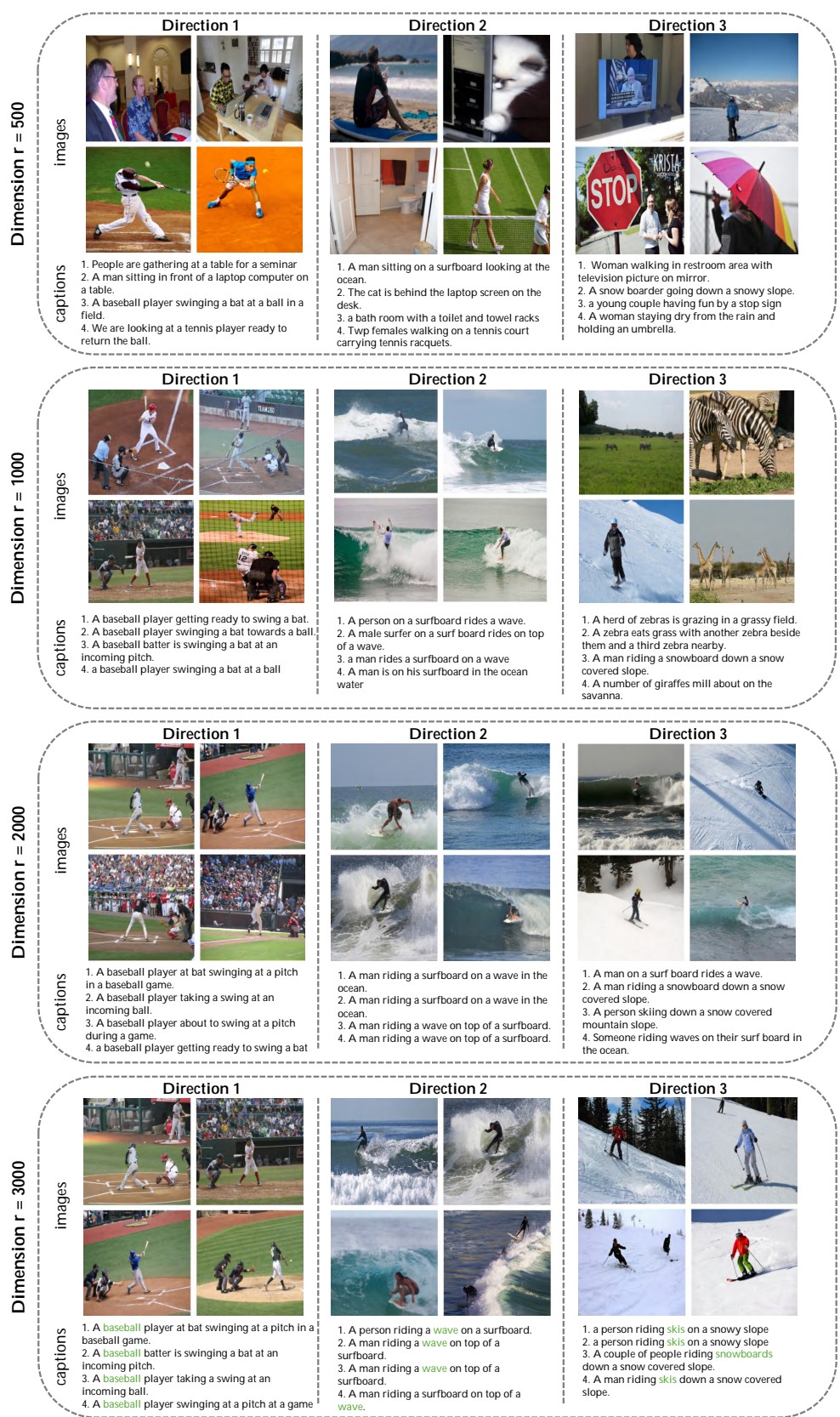

*Figure 19.* Multimodal discrepancy analysis of SigLIP dominant directions relative to OpenCLIP on the MSCOCO dataset under different number of joint random fourier features.

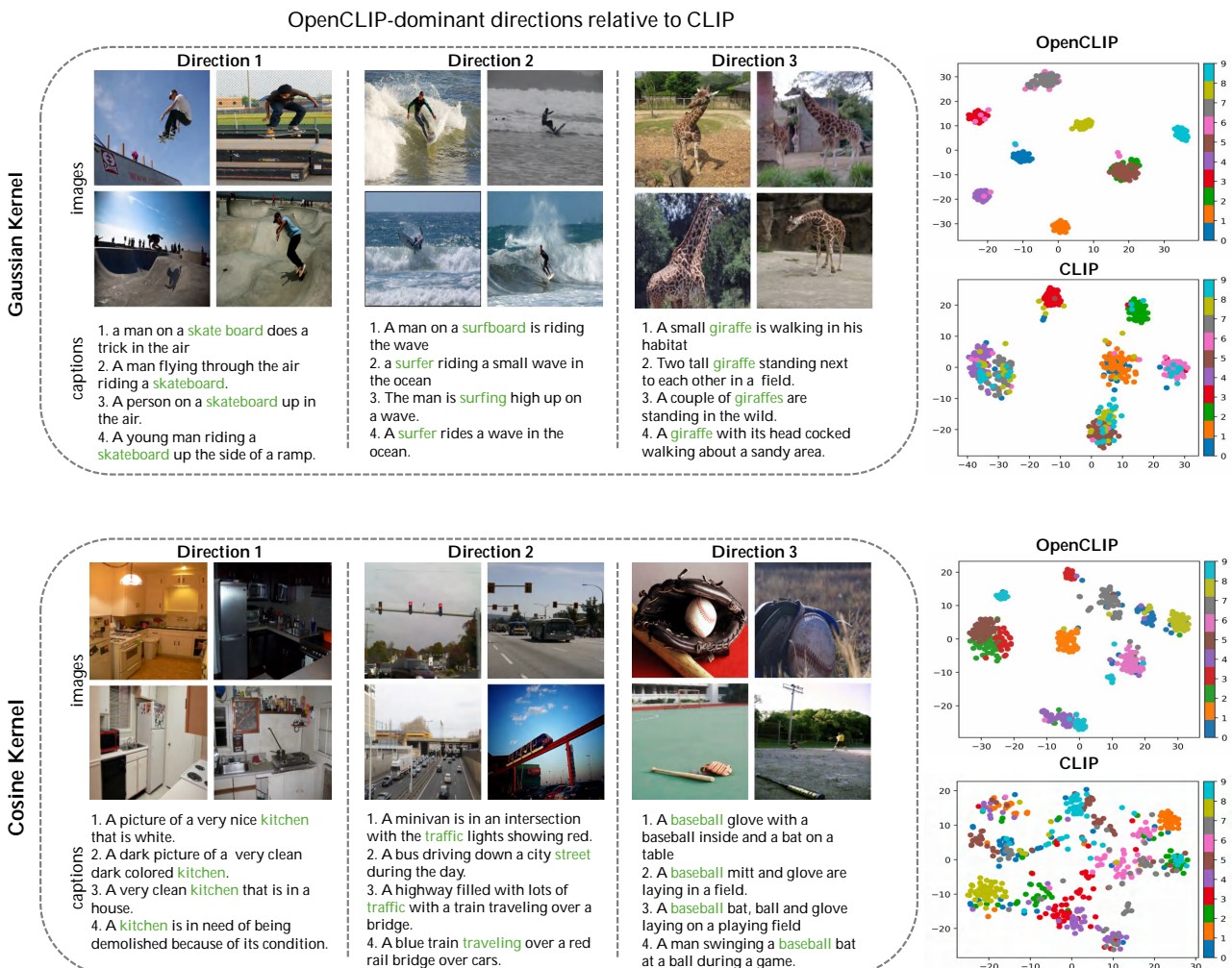

*Figure 20.* Multimodal discrepancy analysis of OpenCLIP dominant directions relative to CLIP on the MSCOCO dataset under gaussian kernel function or cosine kernel function.

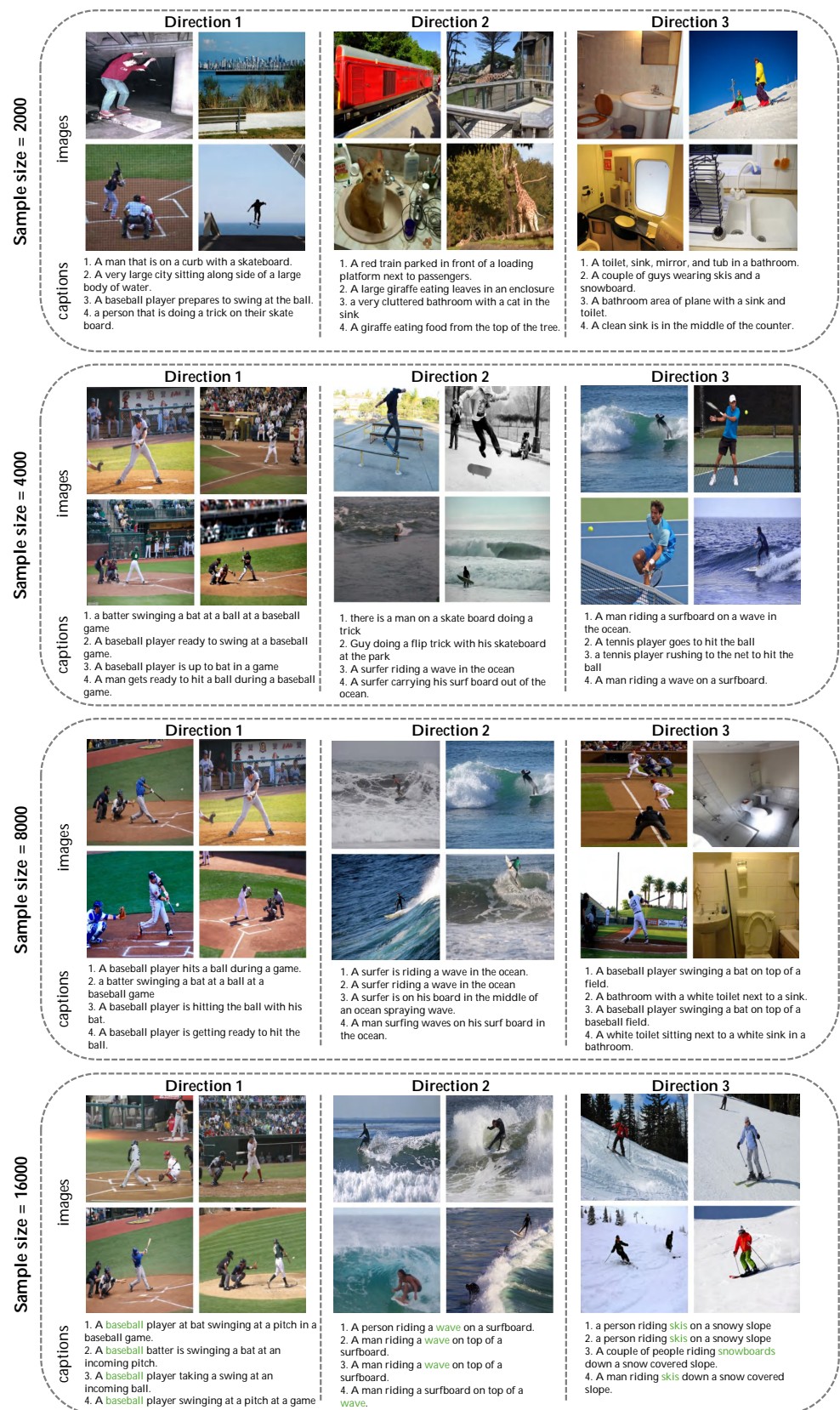

*Figure 21.* Multimodal discrepancy analysis of SigLIP dominant directions relative to OpenCLIP on the MSCOCO dataset under different number of sample size.

