# OpenReview forum: "KODA: Contrastive Representation Comparison and Alignment for Vision-Language Foundation Models"
_ICML.cc/2026/Conference — ICML 2026 regular_

### Official Review · Reviewer_iJPo · 2026-03-03

**Soundness:** 2
**Presentation:** 3
**Significance:** 1
**Originality:** 3
**Overall Recommendation:** 3
**Confidence:** 3

**Summary:**

This paper proposes KODA, a kernel-based framework that reveals where two representation models differ by extracting discrepancy directions that maximize coherence in one model while constraining it in the other via a KKT condition. The method is further extended to multimodal settings through product kernels and joint Random Fourier Feature approximations, supported by theoretical analysis and empirical experiments.

**Compliance With Llm Reviewing Policy:**

Affirmed.

**Final Justification:**

Following the rebuttal process and additiaonl discussions from other reviewers, the evidence and the authors' response of the work have been carefully considered. While the rebuttal clarified several practical aspects, certain concerns persist regarding the empirical significance of the results.

Specifically, in the comparisons provided between aligned-CLIP and models such as DINOv2 or BLIP, the t-SNE visualizations do not exhibit a distinct qualitative advantage, and the numerical gains in metrics like AMI, NMI, and ARI remain difficult to interpret without an absolute baseline for what constitutes a "better" representation in this specific setting. Nevertheless, the authors' point is well-taken that representation-level clustering for interpretability and alignment is a valid and increasingly recognized task, as evidenced by recent work [1].

Although fundamental concerns regarding the framing and the precision of the extracted directions have not been fully resolved, the current trajectory of the research field and the authors' detailed efforts to address the reviewer's comments are respected. Therefore, while the manuscript still requires further refinement in its presentation and motivation in my perspective, the **final score is adjusted to a 3** in recognition of the work's alignment with established research objectives and the emerging standards of the field.

[1] Jalali et al., "Towards an explainable comparison and alignment of feature embeddings", ICML 2025

**Key Questions For Authors:**

See the weaknesses

**Limitations:**

yes

**Strengths And Weaknesses:**

### Strengths

1. The paper is generally well written and easy to follow.
2. The theoretical perspective, using Random Fourier Features to approximate kernels in representation space, is interesting and potentially useful for scalable analysis.
3. The KKT formulation provides a principled way to extract discrepancy directions that highlight differences between two representations.

&nbsp;


### Weaknesses

- The paper’s overall objective and novelty remain somewhat unclear.
  - The method identifies an optimal *discrepancy direction* (a weighting/direction over the reference set) where representation $A$ ($K_A$) shows strong coherence while representation $B$ ($K_B$) is constrained. However, it is not clearly explained what the intended *next step* or *practical outcome* is after discovering such directions/subsets. For example, how can it be used these directions beyond qualitative inspection (e.g., model selection, or improving training objectives)?
  - Additionally, the paper does not sufficiently clarify how to interpret the discrepancy quantitatively. Does a larger discrepancy indicate that representation $B$ is weaker, or simply that the two models organize similarity structure differently? If it is not meant to measure “representational quality,” the paper should explicitly state the intended interpretation and limitations.
- In my view, additional multimodal analysis is necessary to make the experimental conclusions more convincing.
  - On multimodal model results in this paper, it is difficult to interpret what the extracted discrepancy actually reflects. In particular, it is unclear whether the discrepancy arises primarily from **a) a weak or different cross-modal alignment within each model (i.e., how well the image and text encoders are aligned for paired data)?**, or **b) a more general inter-model representational mismatch, similar to the unimodal comparisons.** Without an analysis that separates these two factors, the meaning of the multimodal discrepancies on section 6 and appendices remain ambiguous.
  - The motivation for comparing one model’s dominant directions relative to another is not made sufficiently concrete. The experiments mostly show that such directions can be found, but it is unclear what conclusion should be drawn from these comparisons. This weakens the perceived novelty and purpose of the experimental section.

Minor Weakness:
- Typo: Page 5, line 231-232 $\Rightarrow$ "stpes" $\rightarrow$ "steps"

---

> ### Author Rebuttal · Authors · 2026-03-31
>
> We thank Reviewer iJPo for the thoughtful feedback on our work. Below is our response to the comments and questions in the review:
>
> **The utility and application of KODA**
>
> We would like to provide the following clarifications regarding the practical utility of identifying embeddings differently captured directions in the embedding space.
>
> We first note that our proposed KODA provides a form of representation-level interpretability by identifying which subsets of samples are grouped differently by two embeddings (in terms of the kernel Rayleigh quotient), rather than only reporting an aggregate performance gap. For example, in the unimodal CLIP vs. DINOv2 comparison on FFHQ, KODA identifies fine-grained visual slices, such as samples involving sunglasses, graduation caps, or microphones, that are more clearly separated by DINOv2 than by CLIP (Figure 2 in the main paper). To turn this into an algorithmic design to solve a practical task, we follow the embedding-alignment framework of [1] and fine-tune CLIP on the KODA-identified samples to align it with DINOv2. The resulting kernel heatmaps and t-SNE plots show that the fine-tuned CLIP becomes substantially closer to DINOv2 ([Figure 1](https://github.com/ICML2026Submission9186/ICML2026_Submission9186)). Quantitatively, for each discovered direction, we select 50 representative samples from the corresponding KODA-identified slice, treat the KODA-discovered grouping on that slice as the reference grouping. We then report averaged clustering scores: AMI, NMI, and ARI among 50 runs:
>
> | Model | AMI | NMI | ARI |
> |---|---:|---:|---:|
> | CLIP | 0.25 ± 0.002 | 0.26 ± 0.002 | 0.19 ± 0.001 |
> | finetuned\_CLIP | 0.78 ± 0.004 | 0.78 ± 0.004 | 0.70 ± 0.012 |
> | DINOv2 | 0.83 ± 0.006 | 0.84 ± 0.006 | 0.77 ± 0.017 |
>
> We observe the same pattern in the multimodal setting. On MSCOCO, KODA identifies paired image-caption samples on which BLIP forms a clearer joint structure than CLIP; a representative example is the sports-related slice in Figure 1 of the main text, where tennis-player and baseball-player scenes are clearly separated by BLIP but remain mixed under CLIP. Using these KODA-identified samples, we align CLIP to BLIP and evaluate the same slice again. The aligned CLIP becomes much closer to BLIP in the t-SNE visualizations ([Figure 2](https://github.com/ICML2026Submission9186/ICML2026_Submission9186)) and we also report averaged clustering scores: AMI, NMI, and ARI among 50 runs:
>
> | Model | AMI | NMI | ARI |
> |---|---:|---:|---:|
> | CLIP | 0.52 ± 0.003 | 0.53 ± 0.009 | 0.37 ± 0.004 |
> | finetuned\_CLIP | 0.91 ± 0.006 | 0.91 ± 0.008 | 0.90 ± 0.003 |
> | BLIP | 0.96 ± 0.008 | 0.96 ± 0.005 | 0.96 ± 0.006 |
>
>
>
>
> **Whether the multimodal discrepancy mainly reflects weak/different within-model cross-modal alignment, or a more general inter-model representational mismatch.**
>
> Thank you for this important comment. Our additional analyses suggest that the former plays a major role.
>
> First, we compared discrepancy directions obtained from two multimodal models using only their image embeddings versus using their joint image-text embeddings. The difference is substantial. When KODA is applied to image embeddings only, the discovered directions are much noisier, and the corresponding t-SNE visualizations show weaker and less semantically coherent clustering. In contrast, when KODA is applied to the joint image-text embeddings, the discovered directions align much better with meaningful semantic clusters and the t-SNE structure becomes substantially more concentrated. This suggests that the multimodal discrepancy is strongly tied to cross-modal alignment, rather than being only inherited from unimodal inter-model differences.  The visualization results of such comparison are shown in [Figure 4](https://github.com/ICML2026Submission9186/ICML2026_Submission9186)
> and [Figure 5](https://github.com/ICML2026Submission9186/ICML2026_Submission9186).
>
> Second, to better interpret the multimodal discrepancy, we evaluated image-to-text and text-to-image retrieval on the KODA-selected queries against the full MSCOCO retrieval pool. We found that the selected slice amplifies the gap between SigLIP and CLIP relative to the full set.
>
> | Model | Evaluation set | I2T R@1 (%) | I2T R@5 (%) | I2T R@10 (%) | T2I R@1 (%) | T2I R@5 (%) | T2I R@10 (%) | Avg. gap vs. full |
> |---|---|---:|---:|---:|---:|---:|---:|---:|
> | CLIP | Full | 32.64 | 57.88 | 68.10 | 28.60 | 53.04 | 64.46 | 0.00 |
> | CLIP | KODA-selected subset | 18.00 | 44.00 | 55.00 | 19.00 | 46.00 | 56.00 | -11.12 |
> | SigLIP | Full | 42.64 | 68.32 | 77.98 | 41.86 | 66.42 | 76.22 | 0.00 |
> | SigLIP | KODA-selected subset | 34.00 | 68.00 | 78.00 | 35.00 | 62.00 | 73.00 | -3.91 |
>
> **Typos**
>
> Thank you for pointing out the typo on Page 5, lines 231–232. We will correct “stpes” to “steps” in the revised version.
>
> [1] Gong et al., Kernel-based Unsupervised Embedding Alignment for Enhanced Visual Representation in Vision-language Models. ICML 2025

---

> > ### Author Rebuttal · Reviewer_iJPo · 2026-04-01
> >
> > Thanks to the authors for providing additional results and detailed responses. While the rebuttal improves the paper, my main concerns still remain. In particular, it is still unclear why this approach is fundamentally required and how its intended use should be understood in practice. If KODA is primarily proposed as a tool for interpretation, the paper should explain more clearly what this interpretation is for and what concrete conclusions users are expected to draw from it. For example, is it meant to support model selection, targeted debugging, data curation, failure analysis, or the design of improved training objectives? At present, the paper shows that discrepancy directions can be identified, but the necessity of this analysis and its practical role are still not sufficiently specified.
> >
> > More broadly, the comparison criterion itself remains somewhat ambiguous. It is still unclear whether KODA is meant to be used as a general comparison metric between representations, and if so, under what setting and for what purpose its interpretation should be trusted. Relatedly, the responses still do not clearly separate discrepancy from representational quality, which makes the meaning of the extracted directions less precise. From my perspective, the presentation and framing of the paper are also as important as the novelty of the contribution. Given this, if interpretation is indeed the main contribution, then the overall presentation of the paper should be revised to focus more explicitly on that goal, since the current presentation still feels somewhat vague. For these reasons, I believe the paper would require a fairly substantial, likely major, revision in presentation and motivation. Therefore, I will maintain my score.
> >
> >
> > ========Update========
> >
> > Dear Authors,
> >
> > Thanks for the additional clarifications of this paper. Accordingly, the score has been raised to 3.
> >
> > Best regards, the Reviewer

---

> > > ### Author Response · Authors · 2026-04-01
> > >
> > > We thank the reviewer for the follow-up. We respectfully disagree with the assertion that the motivation and practical role of KODA are unclear, and we believe this concern stems from two points that were already addressed in our response.
> > >
> > > > *“In particular, it is still unclear why this approach is fundamentally required and how its intended use should be understood in practice.”*
> > >
> > > First, the reviewer’s comment questions the motivation of the task itself, i.e., identifying subsets that are clustered differently by two embeddings. However, this task is not introduced by our work. As we have already discussed in the introduction, the recent ICML 2025 paper [1] explicitly formulates identifying mismatched clusters across embeddings as a core objective for explainable comparison of embeddings.
> > >
> > > In this context, KODA should be understood as providing a more principled and effective formulation of this already established task. Therefore, we believe the reviewer’s concern is not directed only at our submission, but applies equally to [1], where the same task was already proposed and positively reviewed in the ICML community.
> > >
> > > Second, the reviewer’s feedback appears to overlook a key part of our rebuttal regarding practical use. We have already demonstrated a concrete pipeline: using KODA to identify subsets and then fine-tuning on those subsets to align embeddings. In both unimodal (CLIP → DINOv2) and multimodal (CLIP → BLIP) settings, this leads to improvements in alignment and clustering metrics. This shows that KODA is not only interpretive, but enables targeted refinement in practice.
> > >
> > > We therefore believe that both the motivation (as an established problem setting) and the intended use (alignment via KODA-identified subsets) are clearly supported by existing work and our additional experiments. We hope the reviewer will reconsider their assessment in light of these points.
> > >
> > > [1] Jalali et al., "Towards an explainable comparison and alignment of feature embeddings", ICML 2025

---

### Official Review · Reviewer_JLNy · 2026-03-05

**Soundness:** 3
**Presentation:** 3
**Significance:** 3
**Originality:** 3
**Overall Recommendation:** 5
**Confidence:** 4

**Summary:**

This paper proposes KODA, a task-agnostic framework for identifying structural discrepancies between pre-trained model representations. Instead of relying on downstream performance, the method frames discrepancy discovery as a constrained optimization problem in Kernel Hilbert Space: seeking directions that maximize sample similarity in model A while minimizing it in model B. This allows for the sequential extraction of orthogonal "discrepancy directions," which are interpreted by identifying top-weighted samples that reveal specific semantic or modal alignment divergences.

**Compliance With Llm Reviewing Policy:**

Affirmed.

**Final Justification:**

The author resolved most of my concerns, so I've upgraded my score to 5

**Key Questions For Authors:**

See weakness

**Limitations:**

yes

**Strengths And Weaknesses:**

Strengths:

1. The idea of using an optimization-based discrepancy identification method to uncover differences in model representations is somewhat interesting.

2. The theoretical derivation in this paper is highly rigorous, and the article is well-structured and logically organized.

Weaknesses:

1. KODA employs a unified kernel function across different models. However, given that pre-training objectives and geometric properties of representation spaces (such as curvature and density) may vary significantly across models, the use of a single kernel function may lead to unfair evaluations.

2. The paper provides an intuitive explanation by presenting the samples with the highest weights in $x$, but the statistical properties of the $x$ vector itself—such as sparsity and distribution stability—are not sufficiently discussed.

3. If KODAs can be considered to evaluate model performance gaps from a clustering task perspective, what distinguishes such evaluation from using downstream tasks? Is it merely that different models exhibit inconsistent performance on specific clustering tasks?

4. Although KODA has identified differences in cluster structures, there is no direct equivalence between ‘inconsistent clustering performance’ and ‘model cognitive superiority or inferiority’. The paper lacks evidence demonstrating that such clustering discrepancies actually translate into reliability differences or semantic understanding biases in real-world tasks. These identified structural differences—do they reflect underlying cognitive biases within the model, or are they merely artifacts generated by the kernel method when handling different distribution densities? Do the variant samples identified by KODA truly exhibit a significant performance gap in downstream tasks?

---

> ### Author Rebuttal · Authors · 2026-03-31
>
> We thank Reviewer JLNy for the thoughtful feedback on our work. Below is our response to the comments and questions in the review:
>
>
> **The use of a single kernel function may lead to unfair evaluations.**
>
> Following prior work [1,2], we use the same Gaussian kernel across models with an appropriately chosen bandwidth, so that the comparison is made under a shared notion of similarity. We also include a cosine-kernel ablation in Appendix Figure 17. We use the Gaussian kernel in the main experiments because it is a standard choice for capturing local grouping structure, which is the main object of comparison in KODA. We will clarify this in the revision.
>
>
> **The statistical properties of the x vector itself**
>
> We agree that the statistical structure of $x$ should be discussed more explicitly. Our goal is not to make $x$ strictly sparse; rather, $x$ is a normalized discrepancy direction over the full reference set, and we want its dominant mass to lie on a structured subset of samples so that the top-weighted examples are representative.
> We conduct this analysis on the FFHQ dataset using the same experimental setup as in the main paper. We quantify the coefficient distributions of the first 10 discrepancy directions and evaluate their stability across 5 runs. The top 1%, 5%, and 10% largest-magnitude entries of $x$ capture on average $18.4\%$, $44.3\%$, and $59.7\%$ of the total $\ell_2$ energy, respectively. The detailed results are shown in the table below.
> | Direction | Top 1\% | Top 5\% | Top 10\% |
> |:---------:|---------:|--------:|---------:|
> | Top-1     | 15.39  | 42.02 | 59.01  |
> | Top-2     | 19.50  | 47.37 | 62.65  |
> | Top-3     | 16.87  | 46.61 | 63.94  |
> | Top-4     | 11.33  | 33.28 | 49.85  |
> | Top-5     | 20.85  | 51.53 | 66.30  |
>
> We further tested stability across 5 random seeds, where the seeds change both RFF sampling and sample ordering. For each of the first 10 directions, the top-10 highest-weighted samples are identical across all 5 runs.
>
> **Application of KODA to downstream tasks**
>
> We would like to clarify that KODA is not intended to replace downstream evaluation, nor is it merely another clustering benchmark. Rather, KODA is designed to identify **where** two models differ in their internal representation structure, before one commits to a specific downstream task.
> A concrete example comes from our multimodal experiment on MSCOCO. In Figure 1 of the main text, KODA identifies a subset of sports-related image-caption pairs, specifically, samples corresponding to semantically similar but distinct scenes such as *tennis-player* and *baseball-player*. In this subset, BLIP forms a much clearer grouping structure than CLIP: in the corresponding t-SNE visualization, these two groups are much better separated in the BLIP embedding space, while they are substantially mixed in the CLIP embedding space. This already goes beyond a standard downstream score: instead of only saying that one model performs better on average, KODA localizes a concrete semantic slice where CLIP has a weaker multimodal representation than BLIP.
> Regarding the applications of KODA in alignment of embeddings, we kindly refer the reviewer to the first item of our response to Reviewer t8gq.
>
> **Do the variant samples identified by KODA truly exhibit a significant performance gap in downstream tasks?**
>
> To verify that these KODA-identified slices are meaningful beyond internal clustering structure, we further evaluated image-to-text (I2T) and text-to-image (T2I) retrieval on the KODA-selected queries against the full MSCOCO retrieval pool, using a setting where SigLIP is stronger and CLIP is weaker on the selected slice. The results show that the KODA-selected slice indeed amplifies the downstream gap between the two models relative to the full evaluation set:
>
> | Model | Evaluation set | I2T R@1 (%) | I2T R@5 (%) | I2T R@10 (%) | T2I R@1 (%) | T2I R@5 (%) | T2I R@10 (%) | Avg. gap vs. full |
> |---|---|---:|---:|---:|---:|---:|---:|---:|
> | CLIP | Full | 32.64 | 57.88 | 68.10 | 28.60 | 53.04 | 64.46 | 0.00 |
> | CLIP | KODA-selected subset-in-full | 18.00 | 44.00 | 55.00 | 19.00 | 46.00 | 56.00 | -11.12 |
> | SigLIP | Full | 42.64 | 68.32 | 77.98 | 41.86 | 66.42 | 76.22 | 0.00 |
> | SigLIP | KODA-selected subset-in-full | 34.00 | 68.00 | 78.00 | 35.00 | 62.00 | 73.00 | -3.91 |
>
>
>
> [1] Zhang et al., "An interpretable evaluation of entropy-based novelty of generative models.", ICML 2024
>
> [2] Jalali et al., "Towards an explainable comparison and alignment of feature embeddings.", ICML 2025
>
> [3] Gong et al., "Kernel-based Unsupervised Embedding Alignment for Enhanced Visual Representation in Vision-language Models.", ICML 2025

---

> > ### Author Rebuttal · Reviewer_JLNy · 2026-04-02
> >
> > Thank you for the authors' reply. I will keep my positive score.

---

> > > ### Author Response · Authors · 2026-04-05
> > >
> > > We sincerely thank Reviewer JLNy for the constructive suggestions as well as the positive feedback on our response. We will incorporate the reviewer’s points and suggestions into our revised manuscript as discussed in our responses.

---

### Official Review · Reviewer_44Wc · 2026-03-11

**Soundness:** 3
**Presentation:** 2
**Significance:** 2
**Originality:** 3
**Overall Recommendation:** 4
**Confidence:** 3

**Summary:**

A variety of multimodal models have been developed and trained in recent years. However, they all have differences in their embedding structures. To analyze these, this work introduces KODA, a method that probes these differences. This is done by solving a constrained optimization problem using the kernel matrices of these two different models. This is possibly prohibitively expensive, though, as kernel matrices of sufficient size to work well also will incur a substantial computational cost. This work also addresses this by projecting with random Fourier matrices to smaller, more tractable ones. Both the original method and the compressed one have sample-complexity bounds. Experiments visually show these differences, both in the setting of using 2 unimodal models of different modalities and with 2 multimodal models with similar modalities. The former shows how language and visual information can be distinguished with KODA, and the latter shows how different multimodal models do not necessarily encode the same information in the same way.

**Compliance With Llm Reviewing Policy:**

Affirmed.

**Final Justification:**

The addition of the quantitatively measurable experiments on using KODA has improved the significance over the original draft. The method is theoretically sound, and it exposes an interesting direction in understanding the differences regarding multimodal models. The main body is quite dense, affecting clarity, but much of the results and the core optimization are quite clear. The rebuttal addressed my main concerns, and overall, this work advances the study of representation discrepancies for multi-modal models.

**Key Questions For Authors:**

1. Eq 1 (and other formulations) are subject to these two constraints. Is it not possible that there are no feasible solutions when using this method empirically? How is that addressed? Especially with the orthogonality constraint for each new direction, the range control on the Rayleigh quotient for model B might become hard to satisfy.
2. Can the differences between the different models (multimodal, unimodal, etc.) be made clearer? For readers who are less familiar with multimodal literature, it would help significantly to know which models are of which type. For example, the unimodal section on page 7 appeared to be looking at the discrepancies between two unimodal models of different modalities, rather than one multimodal and one unimodal.
3. Are there some settings where KODA can be used beyond interpretation? For example, are there ways to use these discrepancies to detect failures in one multimodal model and not the other? At least a discussion of these future directions would help provide significance/utility to KODA.

Minor typos:
- Line 229, poof -> proof
- Line 247, evert -> every
- When referring to proofs in the appendix, can the associated section be mentioned, rather than just mentioning "Appendix"?

**Limitations:**

yes

**Strengths And Weaknesses:**

Strengths:
- The theory justifying this method appears mostly sound. Many of the resulting theorems and bounds are similar to what's expected from other concentration bounds and perturbation bounds
- The justification of KODA (and the random features variant for more efficient computation) by constraining one kernel while maximizing the Raleigh quotient of the other is clear.
- The provided examples in the figures give some interesting, intuitive insights into the discovered differences between visual and textual features.
- Understanding the differences between the different embeddings within multi-models is very useful in understanding these models. The lightweight version with random Fourier features is scalable, adding to its impact/utility.

Weaknesses:
- Upon the first few reads, it was very unclear if this method was for using two unimodal models of different modalities (or a single multimodal model and looking at the differences of the two components) or if this was for two multimodal models and looking at their discrepancies. It still isn't clear, although it appears to be both at different times.
- The main body was quite mathematically dense. While possibly unavoidable as a result of introducing random Fourier features, it did make understanding sections 4 and 5 difficult at first.
- The results are primarily limited to interpretation. There are no experiments or discussions showing how this discrepancy detection can be used beyond interpretability.

---

> ### Author Rebuttal · Authors · 2026-03-31
>
> We thank Reviewer 44Wc for the thoughtful feedback on our work. Below is our response to the comments and questions in the review:
>
> **Possible feasibility concern**
>
> We agree that infeasibility can occur in principle if the constraint level is chosen too aggressively, especially after adding orthogonality constraints for later directions. For unit-norm $x$, the constraint $x^\top K_B x \le \epsilon$ is a Rayleigh-quotient bound, so feasibility on the current orthogonality-constrained subspace $S$ requires $\epsilon \ge \lambda_{\min}(K_B|_S)$.
>
> In practice, we address this in two ways. First, we set $\epsilon$ using a quantile-based rule from the eigenspectrum of model $B$, rather than choosing it arbitrarily, which keeps the target within a feasible spectral range. Second, we solve the problem via a Lagrangian procedure and explicitly monitor the resulting $x^\top K_B x$ values. We will clarify them in the revision.
>
> **The paper should make the compared model types clearer.**
>
> We note that the paper already specifies the unimodal and multimodal model families in the corresponding experimental sections.
> To clarify: The unimodal experiments do not compare different modalities: they compare two visual encoders, DINOv2 and CLIP, using image embeddings on image-only datasets such as AFHQ, FFHQ, and ImageNet. By contrast, the multimodal experiments compare vision-language models—BLIP, CLIP, OpenCLIP, SigLIP, and SigLIP2—on paired image–text data from MSCOCO, using joint image–text representations to study how these models organize the same image–caption pairs.
>
> **Practical Applications of KODA**
>
> We first note that our proposed KODA provides a form of representation-level interpretability by identifying which subsets of samples are grouped differently by two embeddings (in terms of the kernel Rayleigh quotient), rather than only reporting an aggregate performance gap. For example, in the unimodal CLIP vs. DINOv2 comparison on FFHQ, KODA identifies fine-grained visual slices—such as samples involving sunglasses, graduation caps, or microphones—that are more clearly separated by DINOv2 than by CLIP (Figure 2 in the main paper). To turn this into an algorithmic design to solve a practical task, we follow the embedding-alignment framework of [1] and fine-tune CLIP on the KODA-identified samples to align it with DINOv2. The resulting kernel heatmaps and t-SNE plots show that the fine-tuned CLIP becomes substantially closer to DINOv2 ([Figure 1](https://github.com/ICML2026Submission9186/ICML2026_Submission9186)). Quantitatively, for each discovered direction, we select 50 representative samples from the corresponding KODA-identified slice, treat the KODA-discovered grouping on that slice as the reference grouping. We then report averaged clustering scores: AMI, NMI, and ARI among 50 runs:
>
> | Model | AMI | NMI | ARI |
> |---|---:|---:|---:|
> | CLIP | 0.25 ± 0.002 | 0.26 ± 0.002 | 0.19 ± 0.001 |
> | finetuned\_CLIP | 0.78 ± 0.004 | 0.78 ± 0.004 | 0.70 ± 0.012 |
> | DINOv2 | 0.83 ± 0.006 | 0.84 ± 0.006 | 0.77 ± 0.017 |
>
> We observe the same pattern in the multimodal setting. On MSCOCO, KODA identifies paired image-caption samples on which BLIP forms a clearer joint structure than CLIP; a representative example is the sports-related slice in Figure 1 of the main text, where tennis-player and baseball-player scenes are clearly separated by BLIP but remain mixed under CLIP. Using these KODA-identified samples, we align CLIP to BLIP and evaluate the same slice again. The aligned CLIP becomes much closer to BLIP in the t-SNE visualizations ([Figure 2](https://github.com/ICML2026Submission9186/ICML2026_Submission9186)) and we also report averaged clustering scores: AMI, NMI, and ARI among 50 runs on those KODA identified samples:
>
> | Model | AMI | NMI | ARI |
> |---|---:|---:|---:|
> | CLIP | 0.52 ± 0.003 | 0.53 ± 0.009 | 0.37 ± 0.004 |
> | finetuned\_CLIP | 0.91 ± 0.006 | 0.91 ± 0.008 | 0.90 ± 0.003 |
> | BLIP | 0.96 ± 0.008 | 0.96 ± 0.005 | 0.96 ± 0.006 |
>
> These results clarify the intended outcome of KODA: it identifies model-specific weak slices that can be used for targeted refinement, not only for qualitative inspection.
>
>
> [1] Gong et al., “Kernel-based Unsupervised Embedding Alignment for Enhanced Visual Representation in Vision-language Models”, ICML 2025

---

> > ### Author Rebuttal · Reviewer_44Wc · 2026-04-01
> >
> > Thank you to the authors for the responses. The new results, which improve clustering using multimodal models, provide a clear and practical use of KODA, enhancing the significance of the work. As a result, I will raise my score from a 3 to a 4.

---

> > > ### Author Response · Authors · 2026-04-05
> > >
> > > We sincerely thank the reviewer for the constructive review and positive feedback on our response. We are glad that the additional results on multimodal alignment helped to address the reviewer’s comments and to clarify the practical utility and applications of KODA.

---

### Official Review · Reviewer_t8gq · 2026-03-12

**Soundness:** 3
**Presentation:** 3
**Significance:** 2
**Originality:** 2
**Overall Recommendation:** 4
**Confidence:** 2

**Summary:**

This paper proposes KODA, a kernel-based optimization framework for identifying discrepancies between two embedding models. The goal is to discover subsets of samples that form strong groupings in one embedding space while remaining weakly grouped in another. The authors formulate the problem as a constrained quadratic optimization over kernel similarity matrices and derive an efficient spectral solution. The method is further extended to multi-modal embeddings via product kernels and scaled using random Fourier feature approximations. Experiments on unimodal and multimodal datasets illustrate that KODA can identify clusters where embedding models exhibit divergent grouping behaviors.

**Compliance With Llm Reviewing Policy:**

Affirmed.

**Final Justification:**

Most of my concerns have been addressed, so I have decided to raise my score to 4. In addition, as I am not very familiar with the background of this work, my rating may be considered a low-confidence reference.

**Key Questions For Authors:**

1. Clarify the conceptual and algorithmic differences between KODA and prior embedding comparison methods such as SPEC.
2. Expand the experimental evaluation with stronger baselines and include direct comparisons with existing embedding discrepancy analysis methods such as SPEC across multiple datasets.

**Limitations:**

yes

**Strengths And Weaknesses:**

Strengths

1. The paper studies an interesting and relevant problem: identifying structural differences between embedding models beyond standard evaluation metrics such as classification or retrieval accuracy. This perspective is valuable for better understanding representation behavior across models.

2. The discrepancy discovery problem is formulated as a constrained kernel optimization problem with a clear spectral solution, and the proposed covariance block formulation together with random Fourier feature approximations provides a computationally efficient implementation.

Weaknesses

1. The practical implications of discovering divergent groupings between embeddings remain somewhat unclear. While the method identifies clusters where two models behave differently, the paper provides limited discussion on how these discrepancies can be used in practice, such as model debugging, dataset analysis, or model selection.

2. Several components of the proposed method appear closely related to prior work, particularly SPEC (Jalali et al. (2025)). The descriptions in section 'Preliminaries', the use of covariance block formulation and random Fourier features are sisimilar to SPEC. The paper would benefit from a clearer explanation of the conceptual or algorithmic differences and the advantages of KODA relative to related methods.

3. Although the paper discusses multimodal embeddings, the core formulation of KODA is modality-agnostic and can be directly applied to unimodal embeddings. It is therefore not entirely clear what aspects of the proposed approach specifically address the challenges of multimodal representation comparison beyond the use of product kernels.

4. The experimental evaluation provides useful visualizations but remains somewhat limited in terms of systematic comparisons with prior methods. In particular, comparisons with existing embedding discrepancy analysis methods such as SPEC appear only briefly in the paper and are mainly presented in Fig. 8. More comprehensive comparisons would help better assess the relative effectiveness of the proposed approach.

---

> ### Author Rebuttal · Authors · 2026-03-31
>
> We thank Reviewer t8gq for the thoughtful feedback on our work. Below is our response to the comments and questions in the review:
>
> **Practical Applications of KODA**
>
> We would like to provide the following clarifications regarding the practical utility of identifying embeddings differently captured directions in the embedding space.
>
> We first note that our proposed KODA provides a form of representation-level interpretability by identifying which subsets of samples are grouped differently by two embeddings (in terms of the kernel Rayleigh quotient), rather than only reporting an aggregate performance gap. For example, in the unimodal CLIP vs. DINOv2 comparison on FFHQ, KODA identifies fine-grained visual slices, such as samples involving sunglasses, graduation caps, or microphones, that are more clearly separated by DINOv2 than by CLIP (Figure 2 in the main paper). To turn this into an algorithmic design to solve a practical task, we follow the embedding-alignment framework of [1] and fine-tune CLIP on the KODA-identified samples to align it with DINOv2. The resulting kernel heatmaps and t-SNE plots show that the fine-tuned CLIP becomes substantially closer to DINOv2 ([Figure 1](https://github.com/ICML2026Submission9186/ICML2026_Submission9186)). Quantitatively, for each discovered direction, we select 50 representative samples from the corresponding KODA-identified slice, treat the KODA-discovered grouping on that slice as the reference grouping. We then report averaged clustering scores: AMI, NMI, and ARI among 50 runs:
>
> | Model | AMI | NMI | ARI |
> |---|---:|---:|---:|
> | CLIP | 0.25 ± 0.002 | 0.26 ± 0.002 | 0.19 ± 0.001 |
> | finetuned\_CLIP | 0.78 ± 0.004 | 0.78 ± 0.004 | 0.70 ± 0.012 |
> | DINOv2 | 0.83 ± 0.006 | 0.84 ± 0.006 | 0.77 ± 0.017 |
>
> We observe the same pattern in the multimodal setting. On MSCOCO, KODA identifies paired image-caption samples on which BLIP forms a clearer joint structure than CLIP; a representative example is the sports-related slice in Figure 1 of the main text, where tennis-player and baseball-player scenes are clearly separated by BLIP but remain mixed under CLIP. Using these KODA-identified samples, we align CLIP to BLIP and evaluate the same slice again. The aligned CLIP becomes much closer to BLIP in the t-SNE visualizations ([Figure 2](https://github.com/ICML2026Submission9186/ICML2026_Submission9186)) and we also report averaged clustering scores: AMI, NMI, and ARI among 50 runs on those KODA identified samples:
>
> | Model | AMI | NMI | ARI |
> |---|---:|---:|---:|
> | CLIP | 0.52 ± 0.003 | 0.53 ± 0.009 | 0.37 ± 0.004 |
> | finetuned\_CLIP | 0.91 ± 0.006 | 0.91 ± 0.008 | 0.90 ± 0.003 |
> | BLIP | 0.96 ± 0.008 | 0.96 ± 0.005 | 0.96 ± 0.006 |
>
> These results clarify the intended outcome of KODA: it identifies model-specific weak slices that can be used for targeted refinement, not only for qualitative inspection.
>
>
>
> **The differences between KODA and prior embedding comparison methods SPEC.**
>
> To address the reviewer’s comment on the difference between our proposed KODA and baseline SPEC method, we would like to discuss the following points. We first note that while SPEC follows mostly a heuristic and simply considers the eigen directions of kernel difference matrix $K_1- K_2$, we formulate a principled optimization problem for KODA with two clear objectives: 1) Weak clustering of samples in the reference embedding (the constraint in the optimization), and 2) Maximum clusterability in the target embedding (the objective function in the optimization) . This “strong under one / weak under the other” objective is the main conceptual difference of KODA, and it is not merely "implicitly" enforced by simple kernel subtraction as in SPEC.
>
> Empirically, this difference leads to substantially stronger discrepancy directions. As shown in Figure 8 in the appendix, under the same AFHQ setting, KODA yields much larger generalized Rayleigh quotients than SPEC: at $q=0.1$, KODA’s Top-1 direction is about $20\times$ stronger, and its Top-5 average remains consistently above SPEC across all tested constraint levels.
>
>
> **Extension of KODA to multi-modal embedding comparison**
>
> We clarify that KODA is intended as a unified discrepancy-analysis framework for both unimodal and multimodal embeddings. The multimodal contribution is not a different objective, but a principled extension of the same asymmetric objective to paired image–text data. This extension is also technically nontrivial: exact feature-space implementations become infeasible because the joint feature dimension grows multiplicatively across modalities. We therefore introduce a joint random-feature approximation for the product kernel and provide approximation/eigenspace guarantees for this multimodal setting (Theorem 2).We will make this clearer in the revision.
>
>
> [1] Gong et al., “Kernel-based Unsupervised Embedding Alignment for Enhanced Visual Representation in Vision-language Models”, ICML 2025

---

> > ### Author Rebuttal · Reviewer_t8gq · 2026-04-03
> >
> > Some of my concerns have been addressed, but several issues remain. Therefore, I decide to maintain my current score.
> > 1. The method identifies structural differences between embeddings, but its practical use is still unclear. It remains uncertain how these findings can be used in practice to improve models or guide the use of multimodal embeddings.
> > 2. The paper emphasizes multimodal analysis, but the method is essentially a direct extension from the unimodal case via product kernels, and it is unclear whether it anlyzes cross-modal interactions. In addtion, Theorem 2 cannot be found in the paper.
> > 3. The concern about limited comparison with prior methods (e.g., SPEC) is not directly addressed, and more comprehensive evaluation is still needed.

---

> > > ### Author Response · Authors · 2026-04-05
> > >
> > > We sincerely thank the reviewer for the feedback on our response. Below we address the points raised in the follow-up.
> > >
> > > **1. Practical use of KODA**
> > >
> > > We respectfully disagree with the reviewer’s comment about the practical motivation of KODA. As discussed in the introduction, KODA is a principled optimization-based replacement for the heuristic SPEC method in the recent ICML paper [1] that uses a simple eigendecomposition of a kernel-difference matrix. Since KODA and SPEC address **the same underlying task** of identifying subsets that are represented differently by two embeddings, we believe the practical motivation of KODA is at least as well-grounded as that of the SPEC paper [1].
> > >
> > > Also, the reviewer’s feedback appears to overlook a key part of our response about KODA’s practical use. In our previous response, we have already demonstrated a concrete application pipeline: using KODA to identify subsets and then fine-tuning on those subsets to align embeddings. In both unimodal (CLIP → DINOv2) and multimodal (CLIP → BLIP) settings, this leads to significant improvements in alignment and clustering metrics. This shows that KODA is not only interpretive, but enables targeted refinement in practice.
> > >
> > > **2. “the method is essentially a direct extension from the unimodal case via product kernels”**
> > >
> > > We respectfully disagree, because the reviewer’s statement overlooks a key methodological contribution of our work. While the multimodal extension does use a product kernel, the main challenge is not writing down the joint kernel itself, but making KODA **computationally feasible in that joint space**. The computational cost of KODA in kernel space grows cubically $O(n^3)$ with the reference sample size $n$, and in our setting a stable application of KODA can require hundreds of thousands of reference samples. In this regime, directly solving KODA with the product kernel matrix is **computationally infeasible**.
> > >
> > > Our contribution is to use the duality between Kronecker products in feature space and Hadamard products in kernel space to develop a scalable joint random Fourier feature method for KODA in the multimodal setting. This step is essential for applying KODA to large-scale datasets such as MS-COCO and ImageNet. Without this contribution, the multimodal extension would remain only a formal kernel construction and would not be numerically usable on typical datasets.
> > >
> > > We also clarify that by “Theorem 2” in our earlier response, we meant the second theorem of the paper, namely Theorem 5.1 in the main text. We apologize for this mismatch. Theorem 5.1 gives the approximation and eigenspace guarantee for KODA with the proposed joint Fourier features, which is a central part of the multimodal extension.
> > >
> > > **3. “The concern about limited comparison with prior methods (e.g., SPEC) is not directly addressed, and more comprehensive evaluation is still needed.”**
> > >
> > > We respectfully believe this point was already addressed in our previous response. We had already reported comparisons on AFHQ and ImageNet showing that SPEC often fails to identify subsets that are weakly clustered in the reference embedding, whereas KODA successfully finds subsets that are weakly clustered by the reference embedding and strongly clustered by the target embedding, exactly as intended by the asymmetric formulation.
> > >
> > > To further and fully address the reviewer’s concern, we have now extended the comparison to every image dataset considered in the reference SPEC paper [1]. The additional results for three more datasets, provided in Figures 6, 7, and 8 of [the anonymized GitHub repository](https://github.com/ICML2026Submission9186/ICML2026_Submission9186), show consistently across all tested datasets that KODA improves over SPEC in terms of generalized Rayleigh quotient and clustering AMI. We therefore believe that the comparison with SPEC is now comprehensive relative to the evaluation scope of [1].
> > >
> > > [1] Jalali et al., "Towards an explainable comparison and alignment of feature embeddings", ICML 2025.

---

### Decision · Program_Chairs · 2026-04-30

**Decision:**

Accept (regular)

**Comment:**

This paper proposes a kernel-based framework, named KODA, for identifying discrepancy directions between embedding spaces, which is applicable to both unimodal and multimodal representations. Reviewers generally find the technical formulation sound and acknowledge that the method offers good perspectives on representation discrepancy analysis.

After rebuttal, three reviewers support acceptance (two weak accepts and one accept), and they also noted improvements in clarity, additional experiments, and better demonstration of practical utility. These reviewers highlight that the method provides a potentially useful tool for analyzing differences between models and contributes to an emerging line of work on embedding interpretability and alignment.

However, one reviewer (iJPo) maintains a weak reject, while raising concerns. The main unresolved issue is the interpretation and justification of the discovered discrepancy directions. While the method successfully identifies directions where two embeddings differ, it remains unclear:
1) What practical conclusions should be drawn from these directions beyond qualitative inspection or clustering?
2) Whether the discrepancies reflect representation quality, alignment differences, or simply alternative organizational structures in embedding space.
3) How to quantitatively interpret metrics such as AMI/NMI in the absence of a clear external criterion for “better” representations.

This concern is further elaborated in the reviewer–AC discussion. The reviewer emphasizes that the paper makes a conceptual leap from identifying discrepancies to making claims about representation quality or interpretational consistency, without sufficiently demonstrating this step in external validation (like downstream performance, robustness). The current experiments primarily demonstrate that differences can be detected, but do not fully establish why these differences are scientifically meaningful or actionable. The authors are suggested to make full consideration of these concerns in the future version.

Based on the overall score, comments, rebuttal, and discussion, this paper is technically sound in analyzing embedding discrepancies, and the rebuttal has addressed many concerns related to clarity and evaluation, but some questions remain. Therefore, to me, this paper deserves presenting on ICML, but the authors should take into consideration the weaknesses in the final version.